

# Multi-method assessment of reservoir effects on hydrological droughts in an arid region

**Sally Rangecroft**[1], Anne F. Van Loon[1], Héctor Maureira[2], Koen Verbist[3,4], David M. Hannah[1]

[1] School of Geography, Earth and Environmental Sciences, University of Birmingham, Edgbaston, Birmingham B15 2TT, UK

[2] CAZALAC, Water Center for Arid and Semiarid Zones in Latin America and the Caribbean, Benavente #980, La Serena, IV Región, Chile

[3] UNESCO, Enrique del Piano 2058, Santiago, Providenica, Chile

[4] UNESCO Chair for Eremology, Department of Soil Management, Ghent University, Coupure Links 653, 9000 Gent, Belgium

**Sally Rangecroft:** s.rangecroft@bham.ac.uk



**Abstract**
Increasing pressures on water resources in arid regions have led to their increased management and
construction of dams; however, the impacts of these anthropogenic activities on hydrological
droughts have yet to be incorporated and assessed. Here, the impact of the Santa Juana dam on
hydrological drought characteristics downstream has been analysed in the Huasco basin in northern
Chile. Two different methods of drought analysis, threshold level method and standardised indices,
were applied to observed and modelled data. An upstream-downstream approach was taken for the
observation data, analysing the "disturbed" (post-dam) period and the "undisturbed" (pre-dam)
period to allow for an assessment of the onset of the significant anthropogenic activity on the
hydrological regime. Modelled data from the Water Evaluation And Planning (WEAP) model
generated a naturalised scenario and human-influenced scenario for similar analysis. Our findings
show the characteristics of recent drought events in the basin (1965 – 2013). The reservoir is shown
to help alleviate hydrological droughts by reducing frequency, duration and intensity of drought
events, though it did not alleviate major multi-year drought events. A delay in timing of drought
events has been observed also with the presence of the dam. The reliability of these different
methods and approaches to quantify the impact of the dam are evaluated, with concluding
recommendations that the threshold level method using an undisturbed threshold may be the most
suitable. These findings show an applicable way forward with quantifying the human influence on
hydrological droughts, a method that can be applied elsewhere, and on other human activities.
**1.  Introduction**
Drought is an important natural hazard that can lead to severe environmental and socio-economic
impacts in many regions of the world with losses in agriculture, damages to natural ecosystems and
social disruption (Prudhomme et al., 2014; Vicente-Serrano et al., 2014). Drought is regarded as a
deficit in available water compared to the normal conditions ('normal' based on an average over a
certain period or a defined level), which can be established for a range of variables, such as deficit in
precipitation, soil moisture, streamflow, or groundwater. Classically, most drought definitions
consider drought as a natural phenomenon, with climate variability as the only driver of drought.
However, recently there has been a call to acknowledge and include the anthropogenic influence on
drought, drought processes and propagation (AghaKouchak et al., 2015; Wanders & Wada, 2015;
Van Loon et al., 2016a; Van Loon et al., 2016b). Anthropogenic activities can cause, exacerbate or
alleviate drought situations (Vogel & Drummond, 1993; Van Loon et al., 2016a) through directly and
indirectly affecting natural drought propagation and processes (see figure 1, Van Loon et al., 2016a).
For example, human activities can affect the amount of land surface runoff and infiltration (e.g. land
use practices, urbanization, deforestation), water availability (e.g. water abstraction, agriculture/
irrigation) and water storage (e.g. reservoirs). In recent years research has started to incorporate
and investigate the anthropogenic impact on drought (e.g. Wada et al., 2013; Van Loon & Van Lanen,
2013; Mehran et al., 2015; Wanders & Wada, 2015; Liu et al., 2016; Van Loon et al., 2016a; Van Loon
et al., 2016b).
Globally, Wander and Wada (2015) found that drought duration, deficit and intensity were all
worsened by human activity (e.g. water abstractions) through a comparison of the scenarios for the
pristine and human-influenced situation. A limited number of publications have quantified the
human impact on hydrological droughts in case studies through a comparison of the naturalised
situation with the actual. In a few European basins, the abstraction of groundwater on the


hydrological system has been found to result in worse drought impacts than naturally expected or
than meteorological drought impacts (Van Loon & Van Lanen, 2013; 2015). Similarly in China,
hydrological droughts durations and deficits were amplified with the presence of human
disturbances (Liu et al., 2016).
These studies have demonstrated that in general, human water use and activities increases
drought duration and severity; however, this effect can be (partly) compensated by reservoir
regulations that release stored water during the dry period (Wanders & Wada, 2015). Therefore, it is
important to note that human activities (such as reservoirs) can positively affect the hydrological
system through an increased storage capacity, helping with alleviation and resilience during drought
conditions (Mehran et al., 2015; AghaKouchak et al., 2016), reducing the impact of drought through
a change in the timing of water availability, increasing availability during the dry season (Wanders &
Wada, 2015). Flow regulations due to dams and reservoir management are known to be the largest
cause of hydrological alteration (Petts & Gurnell, 2005). However, detailed research on the impact of
dams to downstream drought characteristics such as frequency, timing, duration and intensity are
limited. It is important to fully understand the impact of this human activity and management on the
hydrological system to improve our resilience and adaptation/response to drought.
In arid and semi-arid regions where water availability is mainly supplied by upstream
mountainous areas (e.g. stored as snow and glaciers) or from precipitation in limited periods of the
year, reservoirs are extremely important for water resource management, especially during periods
of meteorological drought. Drought can have large negative consequences in arid and semi-arid
regions and countries due to the high demand for the available resources and the low resilience in
these regions. Although Chile is climatically very diverse, it is a country that suffers from multi-year
droughts. An increase in frequency and severity of drought with a changing climate is projected for
Chile and across the rest of South America (Magrin et al., 2014; WRI Aqueduct, 2014) with negative
impacts associated. In this study we focus on the north of Chile where agriculture is an important
livelihood, despite the extremely arid climate. With increases in demand from population changes
and associated food and water security, and changes in supply through temperature increases and
alterations of precipitation patterns, there are increasing pressures on finite water resources and
their management (Meza, 2013; Rangecroft et al., 2013).
Therefore, there is a need to improve our knowledge on how human activities are impacting on
drought to enable better drought preparation and mitigation, especially in these vulnerable, arid
regions. It is currently unclear on what is the best method for assessing and quantifying the impact
of human activities on hydrological droughts. Subsequently, to address these research gaps the aim
of this paper is to assess the impacts of anthropogenic activity (i.e. dam impoundment and reservoir
storage) on hydrological drought using long-term observations (1965-2013) and model simulations.
This is done using the case study of the Santa Juana dam (built by 1998) in the Huasco basin,
Northern Chile, analysing the impact of this recent human activity on drought occurrence and
characteristics downstream. Through this case study, we test the utility of two different methods of
analysis (standardised indices and threshold level) to find the most appropriate method for
identifying and quantifying the human 'component' of hydrological drought.
**2.  Study area**
**2.1 Huasco Basin**
The Huasco River catchment lies at the limit of the extremely arid Atacama Desert in the north of
Chile (28 – 29 °S) (Figure 1). The Huasco catchment covers 9,850 km$^2$ and the altitude in the basin



ranges from sea level to 5,200 m above sea level (asl). Here, we focus on the upper-mid section of
basin were the dam is located (28 °S, 70 °W) (indicated with a blue triangle on Figure 1). The Huasco
Valley hosts a population of 255,000 inhabitants (Basin-info, 2014), and as in many other semi-arid
regions in the world, the population of the valley relies on the water resources from the upper
catchments in high altitude areas (Viviroli et al., 2007). Although there is limited glacier extent,
glacial meltwaters combined with snowmelt are an important component of the hydrological
resources (Favier et al., 2009). Annual glacial melt can contribute up to 23% of streamflow in the
basin, providing vital water for the regional economy (Nicholson et al., 2009; Gascoin et al., 2011),
with agriculture acting as the main water consumer (85% of total).

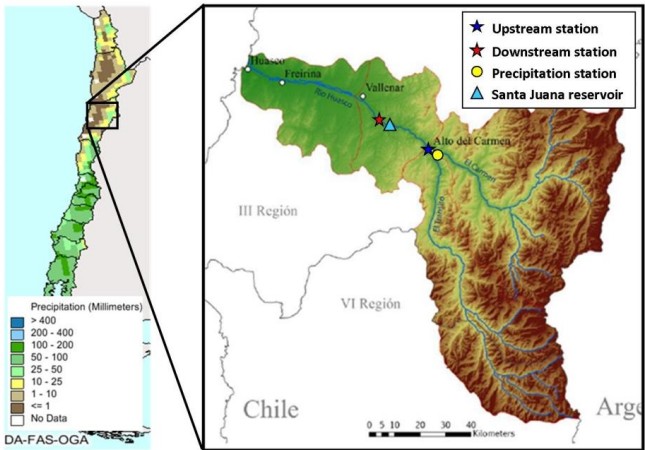


**Figure 1:** Huasco basin: a) identified on a map of annual precipitation of Chile (DA-FAS-OGA); b) topographical
map of the basin (Wagnitz et al., 2014; Fig.1).

The region experiences wet and dry seasons with 80% of annual precipitation occurring

during the wet season, Chilean winter (May – August) (Figure 2). The precipitation is inter-annually
variable with the El Niño-Southern Oscillation (ENSO) (Montecinos et al., 2000; Gascoin et al., 2011).
During El Niño events, positive rainfall anomalies can be observed, whereas below normal conditions
are more likely to occur during La Niña (Verbist et al., 2010; Meza, 2013; Robertson et al., 2014).
Winter periods are most vulnerable to these anomalies. Throughout the basin precipitation is
unevenly distributed, showing a clear altitude gradient with more precipitation occurring in the
mountains. In the basin, precipitation occurs almost exclusively as snowfall, and it is not uncommon
for stations in the lower valley to receive no precipitation in a given year. Peak precipitation (mid-
winter, July) is not seen directly in the discharge data, suggesting the dominance of glacier and
snowmelt in the discharge (peak observed in early summer, December) (Figure 2).



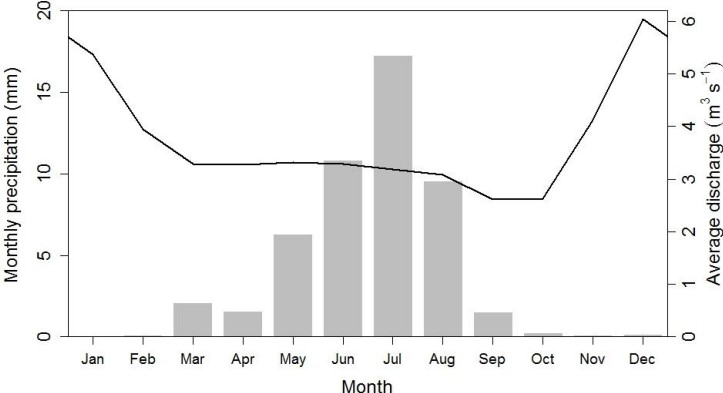

**Figure 2:** Seasonality plots for monthly precipitation and discharge using daily data (1965-2013).

## 2.2 Huasco and water management

An increasing water demand from different water sectors (agriculture, mining, and domestic water
usage) has increased pressure on available water (Gascoin et al., 2011) and its management. The
Santa Juana dam, with a capacity of 170 Mm$^3$, was built in the basin in 1995 and was in operation by
1998. It is the main regulating water structure in the basin, with the purpose of increasing irrigation
security for downstream users. The Huasco basin has been recently managed (2005 – 2015) by the
*"Junta de Vigilancia de Río Huasco y sus Afluentes"* ("Huasco River and its tributaries surveillance
board"), which is monitoring and modelling to calculate water allocations/restrictions. Recent
management and restrictions have been established with the objective to limit impacts of
hydrological droughts across the basin. Regulations on water use depend upon the reservoir levels
(Figure 3a). During the recent multi-year drought (2007-2015), by 2011 reservoir levels dropped
below levels of "partial failure" (<100 Mm$^3$, Figure 3) resulting in the Huasco River Supervisory Board
implementing severe water restrictions (through its operational model).

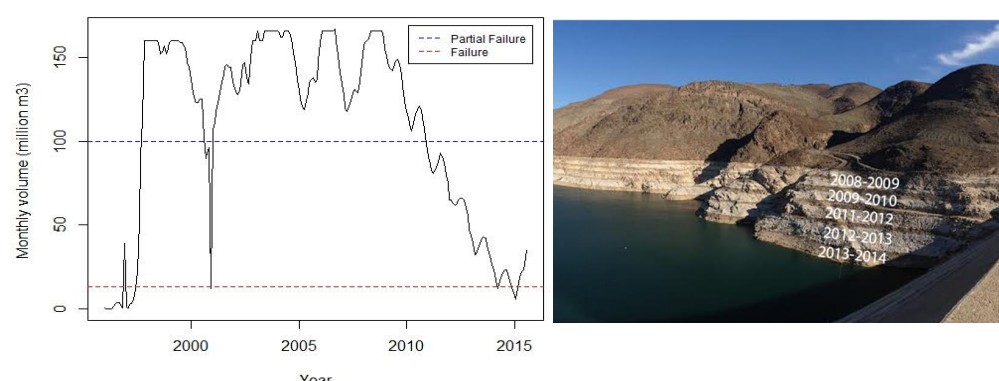


**Figure 3:** The Santa Juana dam in the Huasco basin, built by 1998: a) reservoir levels with failure thresholds; b)
Photograph of the reservoir levels during the recent multi-year drought in Chile starting in 2007 (photo taken
May 2014, Huasco Departmento Técnico report, 2014 Figure 7, page 42).



## 3. Methods

In this study, both observation streamflow discharge data ($Q_{obs}$) and modelled discharge data ($Q_{sim}$) were analysed for drought events and characteristics. An upstream-downstream approach was used on the $Q_{obs}$ to directly compare undistubed (pre-dam) and disturbed (post-dam) data from the presence of the Santa Juana dam (Section 3.1). By using a station upstream and downstream of the dam, a direct comparison about the changes occurring between the two stations can be made (e.g. similar to López-Moreno et al., 2009; Wu et al., 2009). Based on existing ideas and methods, we use the upstream-downstream approach as a new approach for quantifying change (Section 3.4). The dam started operation by 1998, therefore this was a break point dividing the time series into pre- and post-dam periods for the analysis. Modelled data was used to compare simulated discharge data from a naturalised scenario (without the dam, $Q_{sim-nat}$) and a human-influenced scenario (with the dam, $Q_{sim-hum}$) at the same station, downstream (Section 3.2). Two different drought analysis methods were implemented on both types of data: the threshold level (TL) method (Section 3.3.1) and standardised indices (SI) (Section 3.3.2). A comparison of the different methods and data enabled an assessment of the results quantifying the impact of the dam on hydrological droughts downstream (Section 3.4). Data analysis was conducted in the open-source software R using packages including SCI, xts, HydroTSM, and hydroGOF.

### 3.1 Observation data

The observation data for this study were daily precipitation and discharge covering the time period of 1965 – 2013 (Table 1). Data was obtained from the Chilean government's General Water Direction (Dirección General de Aguas, DGA). Missing data was less than 5%, and missing data for hydrological records were replaced by linear interpolation (Hisdal et al., 2004), whereas missing data for the meteorological record were replaced with zeros (Jolly & Running, 2004). Available station data for analysis were limited by their data quality and time lengths. One precipitation station was analysed for meteorological droughts and two discharge stations for hydrological droughts, one upstream and one downstream of the Santa Juana dam (Table 1; Figure 1). However, the focus of this study is on hydrological droughts.

An upstream-downstream approach was used to make a direct comparison between hydrological drought events at both stations (Figure 4). It is known that the exploration of streamflow discharges and the severity and frequency of hydrological droughts in upstream ($Q_{obs-up}$) and downstream ($Q_{obs-down}$) stations is helpful to the understanding of the influence of human activities and its consequences during low-flow periods (López-Moreno et al., 2009; Wu et al., 2009). A baseline period ("undisturbed") from 1965 to 1997 was used as a reference period to indicate the situation before the onset of the significant anthropogenic alternation of the flow, the introduction of the dam by 1998. This is a similar approach to Wang et al. (2009) and Liu et al. (2016). The undisturbed period can also be considered as the "natural" situation, whereas the disturbed period downstream can be considered as the "human-influenced" situation.

### 3.2 Modelled data

The modelled data used here consisted of simulated monthly discharge at the downstream station generated using the Water Evaluation And Planning (WEAP) model for two scenarios (1960 – 2010): 1) "naturalised" with no dam present (also known as "pristine" or "undisturbed" scenario), and 2) "human-influenced" where the Santa Juana dam was present throughout the simulation period. This method is based on the observation-modelling framework presented by Van Loon and Van Lanen



(2013). The modelled discharge was used as input for the drought analysis (Figure 4), which was at
the monthly timescale. The naturalised scenario in the basin allowed a comparison with the human-
influenced scenario, with the only difference being the dam, therefore providing a direct assessment
of the human activity.
The WEAP model is a well-used tool for integrating water resources planning (e.g. Purkey et
al., 2008; Mutiga et al., 2010; Mounir et al., 2011). The model was set up for the Huasco basin using
50 years of historical data as inputs: monthly mean temperature, discharge and precipitation data
from the DGA. The model includes the water demands of the agricultural, industrial, urban and
mining sectors in the basin. WEAP model accuracy was assessed through the Nash-Sutcliffe model
efficiency coefficient (NSE), commonly used to assess the predictive power of hydrological models
(Nash & Sutcliffe, 1970). Monthly $\log Q_{obs}$ and $\log Q_{sim}$ were compared to see how well the model
reproduced observed flow, where a maximum value of +1 indicates a perfect match between model
and observations.


**Figure 4:** Flow diagram to illustrate the data (yellow) and methods of data analysis (orange) used in this study
to then allow for a comparison of hydrological drought characteristics (red).

**3.3 Drought analysis approaches**
Drought events and characteristics were identified and assessed using the common threshold level
(TL) method, or through the standardisation of data into regularly used drought indices,
standardised indices (SI, Tallaksen & Van Lanen, 2004; Vicente-Serrano et al., 2004; Van Loon, 2015).
Drought event characteristics such as timing, duration and severity were extracted from both the
$Q_{obs}$ and $Q_{sim}$ using the SI and the TL methods (Figure 4). The term 'severity' is used throughout to
refer to the deficit volumes produced by the TL method. However, as SI methods do not generate
deficit volumes, severity here also refers to the maximum intensity values produced by SI methods.



### 3.3.1 Threshold level

Drought events can be identified as periods during which the data (river flow, precipitation, etc) are below a certain threshold, known as threshold level (TL) method (Yevjevich, 1967). The TL method is a frequently applied quantitative drought definition. Thresholds based on percentiles of the flow duration curve are commonly employed, with a recommended threshold of between the 70th and 90th percentile for a daily or monthly time series (Van Loon, 2015). The 80th percentile ($Q_{80}$) is frequently used as the threshold for determining a drought (Hisdal & Tallaksen, 2000; Fleig et al., 2006; Heudorfer & Stahl, 2016). $Q_{80}$ is derived from the flow duration curve and is the streamflow value which is equalled or exceeded for 80% of the time. In semi-arid/arid regions, $Q_{50}$ can be used to avoid threshold values of zero (Van Huijgevoort et al., 2012; Giannikopoulou et al., 2014).

When using the TL method, a fixed or variable threshold can be applied to study the deviation from normal runoff, the anomalies (Tallaksen et al., 1997; Hisdal & Tallaksen, 2000; Fleig et al., 2006). The study region has strong seasonality in its precipitation regime (Figure 2) and it is common to find extended dry periods up to several months where no monthly precipitation is observed (Favier et al., 2009; Verbist et al., 2010; Meza, 2013), therefore the variable threshold was used for the meteorological drought analysis. The variable threshold at the 50th percentile was only used on precipitation data to determine meteorological droughts to avoid a threshold of zero. Given the limited seasonality in the discharge data (Figure 2), and because there are no direct comparisons between the precipitation and discharge data, a fixed threshold was used for the hydrological drought analysis, at the 80th percentile.

Droughts can be mutually dependent, where periods of prolonged low discharge are interrupted by short excess periods, which indicates that the system has not had chance to recover from its deficit. Pooling can be applied in order to merge these mutually dependent events and define an independent sequence of droughts (Tallaksen, 2000; Hisdal & Tallaksen, 2000). Here, the inter-event criterion method (Zelenhasić & Salvai, 1987) was used to pool mutually dependent droughts. Optimum inter-event time depends on the regime of the river and the climate of the region (Fleig et al., 2006). The inter-event time period of 15 days was applied based on the sensitivity curve of mean durations at different inter-event time steps that was conducted, as done by Fleig et al. (2006). Minor drought events, which are events of short duration and small deficit volume, can be removed from the analysis using a defined minimum duration. Minor droughts of less than 15 days were excluded from the analysis (as done by Van Loon & Van Lanen, 2013). The TL analysis on monthly data did not require pooling or minor drought events to be dropped as only drought events greater than one month were identified.

Here, the TL method used the "natural", undisturbed period threshold on the human-influenced situation to allow a direct comparison of the impact of the human activity on hydrological droughts. The analysis on the $Q_{obs}$ using the TL method used the pre-dam period upstream data as a reference period to represent the "natural" situation, undisturbed (Figure 4). This was used to calculate the fixed threshold at the 80th percentile, which was then applied to both the upstream and downstream data for the whole time period, as done by Liu et al. (2016). For the TL analysis of the $Q_{sim}$ data, the 80th percentile of the "naturalised" scenario was used as the threshold for both the naturalised and the "human-influenced" situation drought analysis (Figure 4).

### 3.3.2 Standardised Indices

Drought indices are commonly used to assess drought conditions and characteristics based on the measure of deviation from the normal (Stagge et al., 2015) quantifying the number of standard





deviations that an observed value is from the 'normal' value that is calculated over a certain time
period. The Standardised Precipitation Index (SPI) (McKee et al., 1993) is an indicator for
meteorological drought calculated with precipitation data. The process of transforming accumulated
precipitation to the standard normal distribution requires the fitting of a univariate probability
distribution, which is often the gamma distribution, also used here. Through this normalising,
accumulated precipitation can be compared objectively in different climates (Stagge et al., 2015).
The SPI is designed to quantify precipitation deficits on multiple timescales (3 – 48 months). Here,
the SPI is calculated on the 6 month time period (SPI-6) as this is known to be effective in showing
precipitation over distinct seasons (WMO, 2012; Kingston et al., 2015). The SPI over the 12 month
accumulation period (SPI-12) is also used as it is known to perform better in arid climates. Calculated
in a similar manner, the Standardised Streamflow Index (SSI) (also known as the Standardised Runoff
Index, SRI, Shukla & Wood, 2008) is an indicator for hydrological drought using streamflow data
(Svensson et al., 2015; Barker et al., 2016). The SSI uses an accumulation period of 1 month as its
timescale (e.g. Vicente-Serrano et al., 2012; Barker et al., 2016).
The standardised indicator (SI) method standardises each data point according to the time
series norm using a pre-determined threshold to represent drought. For both the SPI and SSI the
commonly used value of -1 was implemented, where values below this represented drought
conditions (McKee et al., 1993; Lloyd-Hughes & Saunders, 2002). The SI method produced monthly
values which were analysed for drought event characteristics (dates, duration, maximum intensity).
SI maximum intensity represented the lowest standardised value of the drought event (Spinoni et
al., 2014) from the data mean (zero).
**3.4 Estimation of the human impact on drought characteristics**
The frequency of drought events, mean and maximum duration, and mean and maximum severity
(deficit or intensity) were obtained through the different drought analysis methods on the $Q_{obs}$ and
$Q_{sim}$. Comparisons are made within and across methods and data. With the $Q_{obs}$ results, a
comparison between the upstream and downstream stations was made, looking at the pre- and
post-dam period (Figure 4). A similar approach has been used by López-Moreno et al. (2009) to study
the transboundary impact of a Spanish/Portuguese reservoir. With the $Q_{sim}$ results, the naturalised
and human-influenced scenarios at the downstream station were compared (Figure 4). For each
drought characteristic, an estimation of the human impact on hydrological drought was estimated
using the following equations.
**3.4.1 Percentage change due to human influence in observation data**
For the $Q_{obs}$ data, to establish the overall percentage change due to human influence with the
upstream-downstream approach, a two stage method was used. In the first step, the natural
difference between the upstream and downstream stations was quantified for the same time period
(Eq 1). These results report the percentage change showing the natural propagation relationship
from upstream to downstream during the pre-dam period, and then the affected propagation during
the post-dam period due to the human influence. Percentages reported are the change downstream
($Q_{down}$) relative to upstream ($Q_{up}$) (Eq 1).
*% of change downstream = [($Q_{down}$ – $Q_{up}$)/ $Q_{up}$] * 100*                    [Equation 1]



In the second step, this natural difference was accounted for and used to establish one overall value
for the percentage of human influence during the human-influence period (post-dam). The pre-dam
period relationship established the percentage difference between the two situations (Eq 1) and was
then used to calculate an expected value for the "natural" situation post-dam downstream, based on
the percentage change and the $Q_{obs-up}$ value post-dam. This generates an expected "natural" value
for the post-dam period which could be directly compared to the actual $Q_{obs-down}$ post-dam value. The
difference between this expected value ($Exp_{hum}$) and the actual observed value ($Obs_{hum}$) gave an
overall percentage of human influence in the post-dam period (Eq 2).

*% of human influence = [($Obs_{hum}$ − $Exp_{hum}$)/ $Exp_{hum}$] * 100*                    [Equation 2]

**3.4.2 Percentage change due to human influence in modelled data**
For the $Q_{sim}$ data, the percentage change due to human influence only needed a direct comparison
of $Q_{sim-nat}$ and $Q_{sim-hum}$ at the downstream station (Eq 3). This could be calculated for the whole time
period, or separated into the pre-dam and post-dam periods for a direct comparison with $Q_{obs}$
results. Percentages reported are the change between the natural and the human situation, relative
to the natural one. $Q_{hum}$ represents human situation, $Q_{nat}$ represents the natural situation.

*% of human influence = [($Q_{hum}$ − $Q_{nat}$)/ $Q_{nat}$] * 100*                    [Equation 3]


4.   **Results and discussion**

**4.1  Drought characteristics in observation data**
Meteorological and hydrological droughts from the observation data were identified using the SI and
TL methods. Visually, the SI results show similar meteorological drought events pre- and post-dam
(Figure 5a), whereas the hydrological droughts are much more severe pre-dam due to the major
event in 1968-1972, known as 'The Great Drought of 1969' in Chile (Figure 5).  Similarly, in the TL
data, hydrological droughts appear much worse in the pre-dam period (Figure 6), and within this
period downstream hydrological droughts seem to propagate into slightly worse events than
upstream (Figure 6b). The differences between meteorological and hydrological droughts reflect the
propagation from meteorological to hydrological drought. However, after completion of the dam by
1998, hydrological drought events appeared to be reduced in their frequency and duration,
especially downstream of the dam (Figure 6). Although similar meteorological droughts occurred
during the post-dam period (e.g. 2004, 2006, 2012), they did not propagate into severe hydrological
droughts during the post-dam period downstream like The Great Drought pre-dam (Figure 5 & 6).
Furthermore, a delay of the drought events can be seen after the building of the dam between the
upstream and downstream SSI (Figure 5b) with droughts downstream occurring later in the year
than upstream, by roughly 8 months. This temporal difference between observed droughts
upstream and downstream reflects the impact of human activities, also observed in other studies
(Assani et al., 2013; Liu et al., 2106).

Quantitative analysis on drought characteristics using SI and TL methods confirms this first

assessment of the results, with nearly all drought characteristics reduced due to the presence of the
dam (% human influence) (Table 2 & 3). The only exception was maximum duration where an
increase (+25%) was seen in the SSI (Table 2). The SSI showed that on average, drought events
downstream were twice as long as upstream pre-dam (+113%), whereas during the post-dam period



they became shorter downstream (-13%) translating into reduction of over a half due to human
influence (-59%) (Table 2). TL method also showed this pattern post-dam (-55%), although drought
durations were already shorter downstream pre-dam in the TL method (-23%), resulting in an overall
estimated decrease of -42% because of human influence (Table 3).
A decrease in average and maximum severity (intensity and deficit) due to introduction of
the dam was seen in both the SSI and TL results, with minor changes seen in the SSI results (Table 2),
but major changes observed in the TL results (Table 3). This was even seen in TL results when
maximum deficit was seen to be larger downstream during the pre-dam period (+28%), yet during
the post-dam period maximum deficit was seen to be reduced largely downstream (-78%), resulting
in an 83% decrease overall due to the human influence (Table 3). Yet, in the SSI, no change was
found between pre- and post-dam in upstream and downstream maximum intensity (Table 2). These
discretions in directions and magnitudes of results are explored further in Section 4.4.
With regards to the meteorological drought, although results show similar events pre- and
post-dam, there are uncertainties regarding the use of standard indicators and regularly applied
methods in regions of limited precipitation. It is important to note that the SPI for meteorological
droughts should be used with caution in arid regions because it has a poor performance with near
zero precipitation (Van Huijgevoort et al., 2012). From our study, we can confirm that SPI is not the
best method to quantify meteorological drought in arid climates like Chile, resulting in unexpected
blocky patterns (Figure 5a). One possible solution to this limitation is to use the consecutive dry
period method (CDPM) (also known as consecutive dry days, CDD), or the newer approach suggested
by Van Huijgevoort et al. (2012) which combines CDD with the variable TL, to disentangle normal dry
periods from drought events.



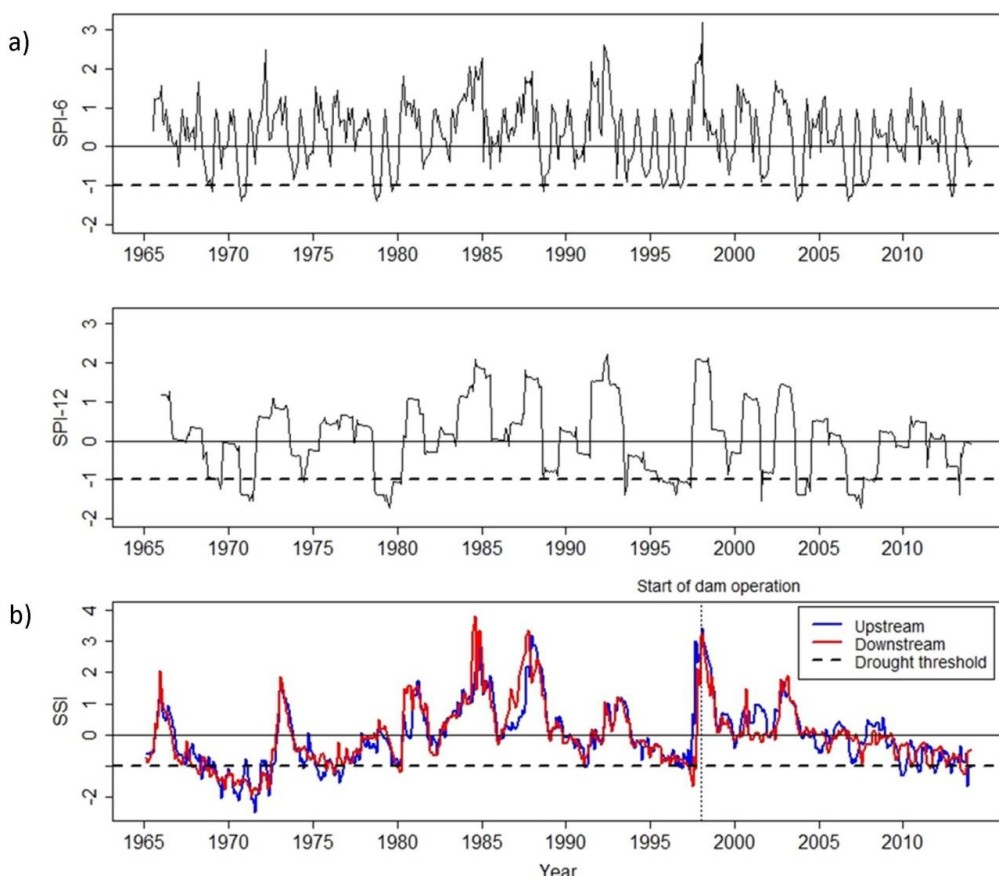


**Figure 5:** Standardised Indices method results for meteorological and hydrological droughts, a) SPI-6 and SPI-12 and b) SSI for the upstream and downstream stations (1965 – 2013). The threshold of -1 was used to identify drought events. The introduction of the dam is indicated on the SSI plot.





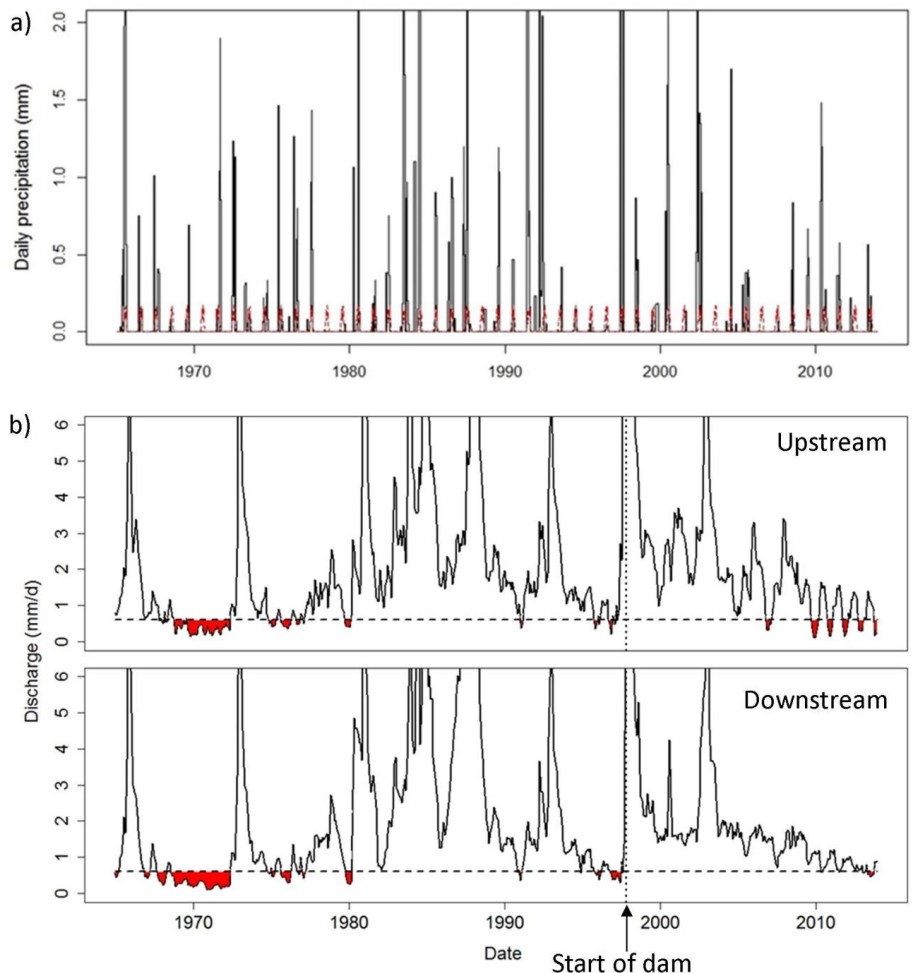

**Figure 6:** Threshold level method results. a) Variable threshold level method ($Q_{50}$) drought analysis on precipitation observation data. b) Fixed threshold level method ($Q_{80}$) drought analysis on upstream and downstream observation data. Monthly data is plotted here for a clearer visual. The upstream $Q_{80}$ threshold is indicated with the horizontal dotted line, and the timing of the dam in operation is highlighted by the vertical dotted line.

### 4.2 Drought characteristics in modelled data

The WEAP model reproduced observed flows well, as indicated by relatively good model efficiency, quantified with the Nash-Sutcliffe model efficiency coefficient, NSE (Nash & Sutcliffe, 1970). The $Q_{obs-down}$ pre-dam period (1965-1997) was compared with the $Q_{sim-nat}$ data for the same time period, giving a "Natural" NSE of 0.817. The $Q_{obs-down}$ post-dam period (1998-2010) was compared with the $Q_{sim-hum}$ data for the same time period, giving a "Human-influenced" NSE of 0.454.

Drought analysis of the WEAP model outputs shows that the presence of the dam changed the characteristics of drought events in the basin. Unlike the $Q_{obs}$ result (section 4.1), the TL method



on the $Q_{sim}$ showed that the WEAP model simulated at least twice as many drought events in the
scenario with a dam than the naturalised scenario (+100%) (Table 4), but a decrease in the rest of
the drought characteristics was seen with this analysis, including a halving of average duration and
deficit volumes with the presence of the dam, but having less reduction on maximum deficit and
duration (Table 4). These results imply that the dam would have more of an impact on reducing
average drought events, but not major ones (shown by less reduction on maximum characteristics
results). SI results show less difference with the presence of the dam, and the results were not in full
agreement with the magnitude and direction of change identified by the TL method. A reduction in
the number of events was seen with the presence of the dam (-20%), and slightly shorter durations
on average (-12%) but no change to the maximum duration, and similar average maximum
intensities (+2%) (Table 4).
One large change in the hydrological system with the presence of the dam is the timing of
discharge peaks and drought events, also observed in the $Q_{obs}$ data (Figure 5 & 7). A delay in the
timing of drought events can be seen between the naturalised and the human-influenced scenarios
(Figure 7). $Q_{sim-nat}$ had only spring and summer droughts, whereas $Q_{sim-hum}$ had mainly winter
droughts. In both, spring and summer hydrological droughts were found in the upstream and
naturalised scenario, whereas winter droughts were mainly seen in the downstream (post-dam) and
human-influenced scenario. This shift in the timing of droughts due to human-influences is an
important deviation from the natural system as it can have implications on ecosystem response and
resilience to drought, especially in arid climates where ecosystems are already sensitive to small
changes in precipitation and available water (Fiebig-Wittmaack et al., 2012). These results are in
agreement with existing studies which found that reservoirs modify the hydrologic regime of rivers,
producing a delay of the natural annual cycle of river and provide a buffering capacity (Petts &
Gurnell, 2005; López-Moreno et al. 2009; Assani et al., 2013) due to increased storage. This
anthropogenic alteration of river regime demonstrates the main management principle of
reservoirs: designed to store water in the wet season and increase water availability for the dry
season, if operated for water supply (Wanders & Wada, 2015), helping build resilience in the system
against drought impacts and hydrological variability (AghaKouchak et al., 2016).



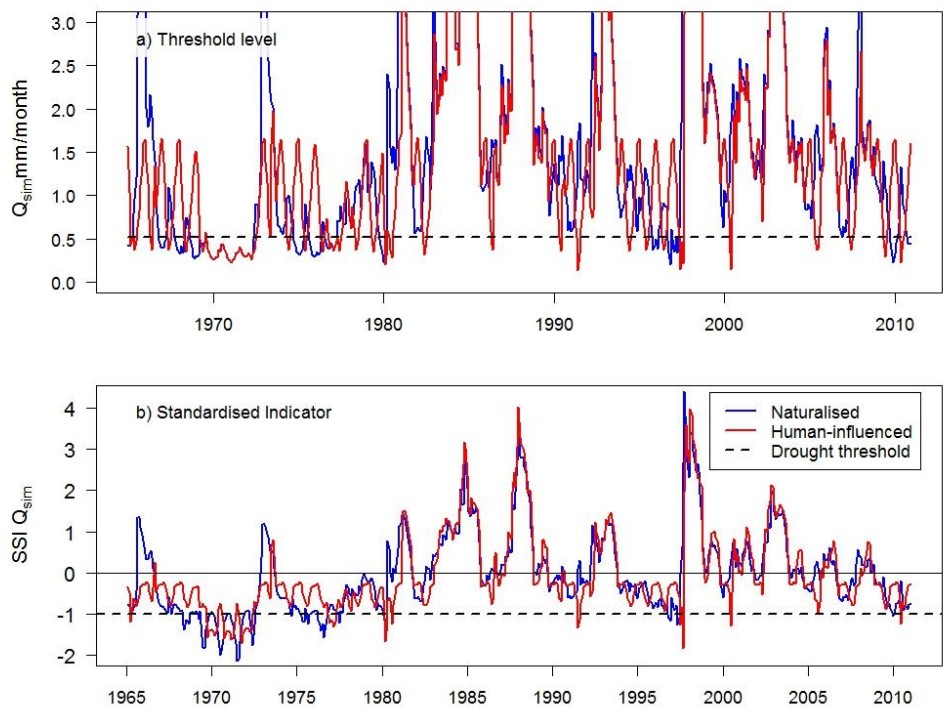

**Figure 7:** Monthly discharge time series for WEAP scenario outputs used for drought analysis with the naturalised (without reservoir, blue) and human-influenced (with reservoir, red) scenarios (1965 – 2010). a) Drought threshold of Q80 for the naturalised scenario is used (dotted line) for the threshold level method drought analysis on both scenarios; b) Standardised indicator (SSI) with the threshold of -1 for both scenarios.

### 4.3 Comparison between observed and modelled data

Overall, a decrease in drought characteristics during the post-dam period was seen downstream in this study (Table 5), but with some disagreements across particular methods and data. Using the direction and magnitude of change calculated amongst the drought characteristics (Tables 2 & 3), the two different methods of drought analysis (SI and TL), and the two different data sources ($Q_{obs}$ and $Q_{sim}$) can be compared for pre-dam and post-dam periods (Table 5 & 6). $Q_{obs}$ gave the same direction of change for half of the drought characteristics results when cross checking the SI and TL methods (5/10 results had the same direction, Table 5), which is more than the $Q_{sim}$ results (3/10 agreement, Table 5). The frequency of drought events in the human-influenced scenario was simulated to be over twice that of the naturalised scenario, which is the opposite pattern to that of $Q_{obs}$ which saw a decrease in drought frequency with the presence of the dam.

The differences between the $Q_{obs}$ and $Q_{sim}$ results could be due to limitations in the WEAP model. Despite the relatively high NSE values, the WEAP model might have issues simulating river flow correctly, especially during the human-influenced period (post-dam) as indicated by the lower



Nash-Sutcliffe value (Human NSE) and the disagreement in the directions of change compared to the
$Q_{obs}$ results (Table 5 & 6). This may be because variations in water use in the basin are not simulated
correctly, which would be especially prominent in the post-dam period where water use restrictions
are implemented based on reservoir levels. These restrictions are not represented in the scenario
modelling, and therefore this may explain the difference between the data and the lower Human
NSE value. Furthermore, other human activities (i.e. demand)in the basin not focused on, such as
land cover and land use change, are likely to have generated some difference between $Q_{obs}$ and $Q_{sim}$
(Liu et al., 2016). The response and feedback of human actions to drought and variable water
availability (such as the regulation of water use dependent upon the reservoir levels) is an important
aspect that needs to be accounted for and further investigated (e.g. Kuil et al., 2016).

**4.4  Comparisons across the different drought analysis methods**
The direction of change was mixed across the drought characteristics for the different methods
(Table 5), but the TL method clearly gave the most agreement on the results between $Q_{obs}$ and $Q_{sim}$
(8/10 same direction, Table 5) compared to SI method (3/10 agreement, Table 5). For the overall
percentage of human influence in the post-dam period, both methods only have less than half of the
results agreeing on the direction of change (2/5 agreeing, Table 6).

An assessment of the application of different methods and data for drought analysis has

been made based on the results of this study. The two different drought analysis methods for
hydrological drought analysis (SI and TL) were conducted here in the way they are most commonly
used in the literature. The results generated with these typical methods consequently differed
substantially (Section 4.3). The SI method had the commonly-used threshold of -1 on monthly data
and included the whole time period to determine drought events, whereas the TL method used the
commonly-applied fixed threshold of the 80[th] percentile on daily data based on a reference period of
"undisturbed" data (1965-1997). Five fundamental differences in the calculations between the two
methods could help to explain the deviation and disagreement in the results seen in this study. The
aim of this paper is not to directly compare between the SI and TL method, but to find the most
appropriate method for analysing the human impact on drought.

Firstly, the SI threshold of -1 is not the equivalent to $Q_{80}$ used in the TL method, therefore

potentially detecting different drought events between the two methods. The 80[th] percentile is
closer to the SI threshold of -0.8. The effect is assumed to be small, as previous studies have found
that changing the threshold level slightly changes the numbers, but not the direction of change (e.g.
Van Loon & Van Lanen, 2013; Heudorfer & Stahl, 2016), however a direct comparison of a -1 and
-0.8 threshold would likely not result in the same number of drought events. Secondly, here a fixed
threshold for the TL was used, whereas the SI is more comparable to a variable threshold because it
calculates the anomaly compared to the climatology of the same month. Heudorfer and Stahl (2016)
recently investigated the impact of a threshold choice on results, finding that it changed the
distribution of drought durations, with a substantial increase seen in the frequency of short droughts
identified by the variable threshold compared to the fixed and a slight decrease in the number of
long droughts.

Third is that the SI method used monthly data, whereas the TL method used daily data. This

may influence the frequency of droughts, their duration and severity as the TL method could identify
shorter droughts. For example, more hydrological drought events were observed using the TL
method than SI (Table 2 & 3), which could be due to the different timescales used. In the TL method,



minimum drought event duration of 15 days was used to remove minor droughts, but the monthly
data used in SI by default had a minimum duration of 30 days.
Fourthly, the SI method does not use a reference period like the TL method, which is based
on a reference period threshold of 1965 – 1997 ("undisturbed"). The advantage of using an
"undisturbed" reference period is that it removes the impact of the dam from the threshold/normal,
whereas the SI method includes the human-influenced data in the calculation of the normal, and
with a different normal different drought events and characteristics are identified (Table 5 & 6).
Furthermore, the fifth difference is that the SI uses station specific data for the threshold,
whereas, TL method used the "natural" station for the threshold to apply to both the "natural"
(upstream) and "human-influenced" (downstream) data. Again, this is useful for directly comparing
the natural situation to the human-influenced. However, using individual stations such as in the SI
method changes the normal against which the droughts are compared, with potentially large
consequences on the results.
**4.5  Sensitivity analysis of drought analysis methods**
These results and discussions have shown that although commonly-used drought analyses have
been applied (SI, TL), the different ways in which it they have been calculated have produced
different results. Therefore, a sensitivity analysis of the aforementioned differences was conducted
(Table 7). For this, we quantified drought with the original SI method using a threshold of -0.8, and
also a modified TL version which uses the variable threshold at the $80^{th}$ percentile with the whole
time period used to calculate the threshold. Station specific thresholds were used in this modified TL
method to simulate the replicate the SI method. The two drought analysis methods were conducted
on the same monthly data for consistency.
The direction of change calculated earlier in this study (Table 5 & 6) can be directly
compared with the results of the sensitivity analysis, with strong agreement found, although some
differences still emerge, mainly in the duration characteristics (Table 7). This was seen more
obviously in the sensitivity analysis for the overall percentage of human influence (Table 8). Looking
across the different methods, the largest disagreement in direction of change can be seen in the TL
method post-dam (Table 7). Therefore it is seen that by recalculating the threshold including human
influences (so same station and whole period), the positive effect of the dam on drought is removed,
implying that this could be happening in the SI method.
It can be seen that the decisions made during the drought analysis process (e.g. daily or
monthly data, fixed or variable threshold, reference period or whole period for the threshold,
upstream/natural or station specific threshold) can affect the results of drought characteristics. This
assessment of the different methods and data suggests that the best approach in this application is
to use is the TL methods on observation data, due to the flexibility of the method to exclude human-
influenced time period from the threshold, and using daily data for finer resolution of results. Using
this suggestion, we can report the $TLQ_{obs}$ results with confidence, a reduction in all drought
characteristics downstream with the presence of the Santa Juana dam, with large reductions in the
average duration and deficit volumes of droughts in the post-dam period (Table 3, 5 & 6). The impact
of the dam was especially seen in drought deficit volumes which showed major reductions
compared to upstream data in the post-dam period and the expected results based on the pre-dam
propagation relationship (Table 3).



### 4.6 Specific major drought events in the basin

Here we focus on two major drought events in the time series to explore the observed impact and modelled impact of the dam on drought characteristics downstream, one during the pre-dam period, The Great Drought (1968-69), and one in the post-dam period, the recent multi-year drought (2007 – 2015).

The Great Drought of 1968-69 was an event with some of the largest deficits across the country during the twentieth century (seen in Figure 6b). The impacts were felt across the region with huge losses for crops (potato, rice, maize, beans), fruit trees, and vineyards, livestock died, and associated milk, meat and wool outputs declined. In the Huasco region farmers and communities also lacked water for human consumption. Across all the methods and data this was represented as the worst hydrological drought in both duration and severity. Drawing on the recommendations from this study, using the TL method for the drought analysis, the $Q_{obs}$ results at the downstream station showed the drought event to last 46 months (drought event observed Sept 1968 – May 1972) and $Q_{sim}$ results showed the event to be of similar duration (45 months) in the naturalised scenario. However, $Q_{sim-hum}$ results suggested that with the presence of the Santa Juana dam, the event would have been alleviated for the first year (drought event modelled Sept 1969 – July 1972). Therefore, the presence of the dam would have helped to alleviate the start of the drought event, however, it would have not prevented against a major drought fully; according to the simulations, the drought would have still persisted for three years.

More recently, in 2007 a multi-year drought started in Chile, hampering copper production (of which Chile is the world's number one exporter), exacerbating forest fires, driving energy prices higher (due to reduced hydro-power production), and negatively impacting agriculture (Boisier et al., 2016). The effects of the drought were intensified by an increasing demand driven by the country's economic growth; Chile's economy has more than doubled in a decade. The quantity of water stored in reservoirs dropped dramatically during the drought (e.g. Santa Juana dam, Figure 3). Again, just focusing on the recommended TL method, this multi-year drought event was represented in the $Q_{obs-up}$ data as series of hydrological events from 2006 until the end of the time period (Figure 6b) whereas in $Q_{obs-down}$ the drought events do not occur until 2010, when reservoir levels entered partial failure (Figure 3). Drought events downstream of the dam are clearly much shorter in duration and deficit until the end of the time series (Figure 6b). These results suggest that the presence of the dam has helped to alleviate the recent multi-year drought in the downstream station, postponing the onset of events for the first four years because of the increased storage in the system.

These results have shown that even though reservoirs are seen to have a positive effect in alleviating droughts, they are often not resilient enough to completely protect against large multi-year droughts, although this is partly related to reservoir size and management. Similar to these results, it has also been suggested through socio-hydrological modelling that reservoirs have been seen to result in less frequent drought impacts, but for major drought events where the reservoirs run dry, drought impacts may be much more severe (Kuil et al., 2016). Therefore, it can be argued that other ways to build resilience against these major droughts, other than just increasing and managing storage, are necessary. For example, a large reduction in water consumption in the Australian city of Melbourne was seen to help alleviate the impacts of the Millennium Drought (Low et al., 2015). However, caution should be applied to coping strategies to drought which involve an over-abstraction of groundwater supplies, as this has been seen to worsen droughts, lowering groundwater levels and lengthening recovery time of the groundwater system (Van Loon & Van Lanen, 2013).


## 5.  Concluding remarks

### 5.1 Impact of reservoirs on hydrological droughts

These results on drought characteristics in the past half a century have not been shown before for this basin, this region, or this topic, therefore providing useful information on drought frequency and characteristics in a vulnerable environment with regard to water resources. This is also the first attempt in the region to quantify the impact of a human activity, the presence of a dam, on hydrological droughts. Overall, in the Huasco basin a decrease in the frequency, duration and severity of drought events was observed downstream of the Santa Juana dam, showing that the presence of a reservoir provides resilience against short-term droughts. However, this study also found that the reservoir could not alleviate fully against major multi-year droughts and therefore it is important to increase resilience in other ways.

A delay in timing of drought events with the presence of the dam was also seen downstream, showing redistribution in water availability solely due to human activities, the regulation of water from the reservoir. It was seen that the reservoir altered the river regime downstream, causing a delay in the timing of hydrological droughts (from spring/summer droughts to winter droughts), which could have an important impacts on ecosystems, especially in sensitive environments such as arid regions. Therefore, it is important to monitor and increase research and understanding of the impact of human activities on the hydrological system, particularly in these semi-arid regions, but also worldwide.

### 5.2  Quantifying human influence on droughts: ways forward

This work highlights the importance of including and assessing the impact of anthropogenic activities into drought analysis. This research has shown how the two methods of analysis used (standardised indices and threshold level) differed, possibly due to the differences between how the different methods include or exclude the human influences in the "normal" situation against which drought is assessed. This work suggests the need for care when choosing data and method for drought analysis, as those decisions are seen here to affect the results. Using an undisturbed reference period, as the threshold level method did here, helps to exclude the human impact from the threshold/ "normal", allowing for more direct conclusions about the human impact during the disturbed period. Whilst the focus of this study was on the impacts of a reservoir on hydrological droughts, other activities such as irrigation, groundwater water abstraction, and urbanisation should also be investigated across different climates, river basins and societal contexts. This work show an effective way forward to quantify the human influence on hydrological droughts, the recommended TL method with an "undisturbed" period for the threshold, that can be applied elsewhere, and on other human activities, to increase our understanding of the impacts of anthropogenic activities on hydrological droughts



**Author contribution:** SR and AVL came up with the concept for the manuscript. SR conducted the analysis. SR wrote the manuscript with the input from all co-authors (AVL, DH, KV, HM). AVL provided continuous input and insight. KV and HM provided access to the data and local information from the basin. The authors would also like to thank the editor for their valuable comments.

**Competing interests:** The authors declare that they have no conflict of interest.

**Acknowledgements:** This project was funded by the Dutch NWO Rubicon Project "Adding the human dimension to drought" (reference number: 2004/08338/ALW). The present work was (partially) developed within the framework of the Panta Rhei Research Initiative of the International Association of Hydrological Sciences (IAHS) (see http://iahs.info/Commissions--W-Groups/Working-Groups/Panta-Rhei/About-Panta-Rhei.do). The authors would like to thank Pablo Rojas and Sergio Alejandro Gutiérrez Valdés at Junta de Vigilancia de Río Huasco y sus Afluentes, Chile for their work on the WEAP model in the Huasco basin. The authors would also like to thank the research group at the University of Birmingham for their discussions and Niko Wanders for his continued support for the ideas and analysis.



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





**Table 1:** Data stations used for analysis, 1965 - 2013 (source: DGA, Chile)

| Variable measured | Location (catchment area) | Station name (number) | Coordinates | Elevation (m) | Annual mean (+/- range) |
|---|---|---|---|---|---|
| Precipitation | Upstream | Junta del Carmen (P3804007) | 28.75731 S 70.48385 W | 770 | 49 mm (0 - 234.5) |
| Discharge | Upstream (1026.7 km$^2$) | Rio transito antes junta rio Carmen (Q3806001) | 28.75369 S 70.48495 W | 812 | 0.077 mm/d (0.0099 − 0.286) |
| Discharge | Downstream (1516.4 km$^2$) | Rio Huasco en Santa Juana (Q3820003) | 28.67536 S 70.64857 W | 553 | 0.074 mm/d (0.0061 − 0.309) |



**Table 2:** SPI and SSI drought characteristics (number of events, average duration and average deficit) using threshold of -1. Values are reported to 1 decimal place (d.p.), except for intensity values which are reported to 2 d.p. Durations are reported in months.

| Standardised Indices | Pre-dam (1965 – 1997) | | | | Post-dam (1998 – 2013) | | | | % human |
| Observation data | SPI | SSI | | % change down stream | SPI | SSI | | % change down stream | influence |
| Q(mm/d) | | Upstream | Down stream | | | Upstream | Down stream | | |
|---|---|---|---|---|---|---|---|---|---|
| Number of events | 7 | 14 | 7 | -50 | 4 | 5 | 4 | -20 | +60 |
| per decade | 2.1 | 4.2 | 2.1 | | 2.7 | 3.1 | 2.5 | | |
| Average duration | 2.3 | 4.4 | 9.3 | +113 | 3.5 | 2.6 | 2.3 | -13 | -59 |
| Maximum duration | 5 | 21 | 46 | +119 | 5 | 4 | 5 | +25 | -43 |
| Average max. intensity | -0.15 | -1.38 | -1.36 | -2 | -0.19 | -1.33 | -1.11 | -17 | -15 |
| Max. intensity | -1.39 | -2.48 | -1.95 | -21 | -1.39 | -1.64 | -1.28 | -22 | -0.5 |


**Table 3**: Drought event characteristics from the upstream and downstream observation data, pre- and post-dam. Percentage change is reported as the change downstream compared to upstream. Drought events were calculated using the TL method with a threshold of $Q_{80}$ for the reference period (1965-1997) from the upstream data, with a pooling of an inter-event criterion < 15 days, and minimum drought event duration > 15 days.

| Threshold Level Observation data Q(mm/d) | Pre-dam | | | | Post-dam | | | | % human influence |
|---|---|---|---|---|---|---|---|---|---|
| | Precip. | Upstream | Down stream | % change down stream | Precip. | Upstream | Down stream | % change down stream | |
| No. of events | 38 | 17 | 18 | +6 | 22 | 7 | 5 | -29 | -33 |
| per decade | 11.5 | 5.2 | 5.5 | | 13.75 | 4.4 | 3.1 | | |
| Average duration | 2 | 8 | 6 | -23 | 2 | 4 | 2 | -55 | -42 |
| Max. duration | 4 | 46 | 47 | +2 | 4 | 6.2 | 4.8 | -22 | -23 |
| Average deficit | 3.95 | 1.35 | 1.28 | -6 | 3.40 | 0.88 | 0.12 | -86 | -86 |
| Max. deficit | 8.78 | 10.53 | 13.54 | +28 | 8.78 | 1.54 | 0.33 | -78 | -83 |



**Table 4:** Drought event characteristics calculated from WEAP modelled monthly data, the natural and human-influenced scenarios (1960 -2010). Percentage change is reported as the change human compared to natural. SI methods used the threshold of -1 to determine a drought event. TL method used a threshold of $Q_{80}$ from the natural scenario as the reference period. Intensity is only reported for SI method and deficit for the TL method.

| Modelled data mm/month (1960 – 2010) | Standardised Indices | | | Threshold Level | | |
|---|---|---|---|---|---|---|
| | Natural | Human | % human influence | Natural | Human | % human influence |
| Number of events | 15 | 12 | **-20** | 14 | 28 | **+100** |
| No. of events per decade | 3.1 | 2.5 | | 2.9 | 5.8 | |
| Average duration | 4.1 | 3.6 | **-12** | 7.9 | 4 | **-47** |
| Maximum duration | 11 | 11 | **0** | 45 | 35 | **-22** |
| Average max. intensity/deficit | - 1.4 | - 1.4 | **+2** | 1.19 | 0.55 | **-54** |
| Maximum intensity/deficit | -2.1 | -1.8 | **-14** | 8.14 | 6.88 | **-15** |





**Table 5:** Changes observed between upstream and downstream stations for $Q_{obs}$ and between natural and human scenarios for $Q_{sim}$ the different methods (SI & TL). A blue symbol represents a decrease in drought characteristics due to the human influence (alleviation); a red symbol represents an increase in drought characteristics due to the human influence (aggravation). A coloured outline represents a minor change (< ±50%) due to human influence, and a block colour symbol represents a major change (> ± 50%). ≡ represents the same as, ≈ means nearly equal to (zero ± 5%).

| Compared methods % change downstream | $Q_{obs}$ | | | | $Q_{sim}$ | | | |
|---|---|---|---|---|---|---|---|---|
| | Pre-dam | | Post-dam | | Pre-dam | | Post-dam | |
| | SI | TL | SI | TL | SI | TL | SI | TL |
| No. of events | ▼ (blue, major) | △ (red, minor) | ▽ (blue, minor) | ▽ (blue, minor) | ▽ (blue, minor) | ▲ (red, major) | ▲ (red, major) | ▲ (red, major) |
| Average duration | ▲ (red, major) | ▽ (blue, minor) | ▽ (blue, minor) | ▼ (blue, major) | ≈ | ▽ (blue, minor) | ≡ | ▽ (blue, minor) |
| Max. duration | ▲ (red, major) | ≈ | △ (red, minor) | ▽ (blue, minor) | ≡ | ▽ (blue, minor) | ▼ (blue, major) | ▽ (blue, minor) |
| Average max. intensity/deficit | ≈ | ▽ (blue, minor) | ▽ (blue, minor) | ▼ (blue, major) | ≈ | ▼ (blue, major) | △ (red, minor) | ▽ (blue, minor) |
| Max. intensity/deficit | ▽ (blue, minor) | △ (red, minor) | ▽ (blue, minor) | ▼ (blue, major) | ▽ (blue, minor) | ▽ (blue, minor) | △ (red, minor) | ▽ (blue, minor) |





**Table 6:** Changes during the post-dam period between the natural and human-influenced situation for the different data, Q$_{obs}$ and Q$_{sim}$ and methods, SI and TL. A blue symbol represents a decrease in drought characteristics due to the human influence (alleviation); a red symbol represents an increase in drought characteristics due to the human influence (aggravation). A coloured outline represents a minor change (< ±50%) due to human influence, and a block colour symbol represents a major change (> ± 50%). ≡ represents the same as, ≈ means nearly equal to (zero ± 5%).

| % human influence post-dam | SI | | TL | |
|---|---|---|---|---|
| | **Q$_{obs}$** | **Q$_{sim}$** | **Q$_{obs}$** | **Q$_{sim}$** |
| No. of events | ▲ | ▲ | ▽ | ▲ |
| Average duration | ▼ | ≡ | ▽ | ▽ |
| Max. duration | ▽ | ▼ | ▽ | ▽ |
| Average severity | ▽ | △ | ▼ | ▽ |
| Max. severity | ≈ | △ | ▼ | ▽ |



**Table 7:** Sensitivity analysis: Percentage change downstream of $Q_{obs}$ data using SI and TL methods (left) with sensitivity analysis results (right). Blue represents a decrease in characteristics, red an increase, with a block colour symbol representing a change greater than ±50%.

| Direction of change downstream | $Q_{obs}$ | | | | $Q_{obs}$ sensitivity analysis | | | |
|---|---|---|---|---|---|---|---|---|
| | Pre-dam | | Post-dam | | Pre-dam | | Post-dam | |
| | SI | TL | SI | TL | SI | TL | SI | TL |
| No. of events | ▼ | △ | ▽ | ▽ | ▽ | △ | ▽ | ▽ |
| Average duration | ▲ | ▽ | ▽ | ▼ | △ | △ | △ | △ |
| Max. duration | ▲ | ≈ | △ | ▽ | ▲ | ≈ | ▽ | ▲ |
| Average max. intensity/deficit | ≈ | ▽ | ▽ | ▼ | ▽ | △ | ▽ | ▽ |
| Max. intensity/deficit | ▽ | △ | ▽ | ▼ | ▽ | △ | ▽ | △ |





**Table 8:** Sensitivity analysis for the percentage of human influence (calculated from observed compared to expected). Blue represents a decrease in characteristics, red an increase, with a block colour symbol representing a change greater than ±50%.

| % human influence | Q_obs | | Sensitivity analysis | |
|---|---|---|---|---|
| | SI | TL | SI | TL |
| No. of events | ▲ (red) | ▽ | ▽ | ▽ |
| Average duration | ▼ (blue) | ▽ | ▽ | △ (red) |
| Maximum duration | ▽ | ▽ | ▼ (blue) | ▲ (red) |
| Average severity | ▽ | ▼ (blue) | ≈ | ▽ |
| Maximum severity | ≈ | ▼ (blue) | ≈ | ≈ |