# Peer review of "Multi-method assessment of reservoir effects on hydrological droughts in an arid region"

_Earth System Dynamics, 2016_

## Referee Comment (RC1) · Anonymous Referee #1 · 19 Dec 2016

This paper study the anthropogenic effect of the Santa Juana dam on drought in the Huasco basin in Chile. The Standardized Indices (SI) and Threshold Level (TL) method with different thresholds are applied to observed and modelled data. The study illustrates the effect of the Santa Juana dam on different characteristics of the drought, including duration and intensity. Also, the performance of the SI and TL method on the observed and simulated data is compared. The topic of this manuscript is interesting, as reviewer found the study was more like to investigate the reservoir operation and reservoir's capability on mitigating drought conditions instead of drought monitoring and prediction. The current manuscript suffers from some major issues as reviewer listed below.

Major comments: 1. Motivation: Reservoirs are able to change the patterns and magnitudes of streamflow. This is an unknown fact. Reviewer is unclear about the experiment

[Figure]

design of this study. If both observed and simulated results shown reservoir was able to delay the timing, frequency or magnitude of drought events, then this again is a re-state of known fact. Therefore, how this study can be used for more efficient reservoir operation or planning for droughts events still have questions. Authors are suggested to summarize more about recent published paper in WRR and GRL to better formulate the motivation of this study. (search key words: reservoir operation, water allocation, drought conditions).

2. Motivation and Organization: The introduction has described why drought is impor-tant and previous studies focusing on the relationship between human activities and drought conditions. However, reviewer find it difficult to understand the contribution of this study with respect to those mentioned studies in this manuscript. What is the scientific problem that this study is trying to solve and how this study will contribute to those existing findings about human activities amplified or alleviated drought condi-tion? Reviewer believe this study is not targeting on methodology advances rather than application of certain existing analytical methods to a case study in Chile. Then, the question become how this application is unique and novel as compared to the uses of similar techniques to analyze the functionality of reservoirs on droughts.

3. Justification of results: In results section (line 337-338), authors also quoted that "This temporal difference between observed droughts upstream and downstream re-flects the impact of human activities, also observed in other studies (Assani et al., 2013; Liu et al., 2106)." Then, the explanation is needed to justify in what aspects this study will differ from others?

4. Justification of results: The "human-influenced" results only reached NSE=0.454, which is a rather poor performance. Considering the temporal resolution (monthly) and the length of analysis (over decade), this NSE value lacks proper accuracy to represent any simulation of human activities. This critically undermines the main motivation of analyzing the functionalities of reservoirs in alleviating drought conditions.

[Figure]

5. Justification of results: line 15, "A delay in timing of drought events has been observed also with the presence of the dam" how is this being illustrated in the results and conclusion as this is one of the main concluding remarks in the end?

6. Experiment Design: Reviewer also found it is confusing about the experimental design. For instance, in Figure 4, both daily data from 1965-2013 for upstream discharge (this becomes reservoir's inflow) and downstream discharge (reservoir's outflow) were used in the "pre-dam versus post-dam" scenario. The results for the first scenario was compared with a "naturalized versus Human-influenced" scenario as shown on the right panel, in which the resolution of data become monthly and coverage of data starts from 1960 to 2010. It is confusing, or lack of explanation why different resolution and lengths of data were used and compared. Isn't that Pre-dam the same as Naturalized, and Post-dam is human-influenced?

7. Experiment Design: Reviewer noticed that the length of the observed data and simulated data are different (Figure 5, 6, and 7). Any comments on some drought events happened after 2010 as many figures are showing a significant decreasing trends during the recent years?

8. Clarity and Presentation: A general overview of the WEAP model (inputs, output, general structure…) can give better understanding of the result section, as the author mentioned variation in water use as the reason for the performance of the model.

9. Support of Conclusion: Line 575-592. This section again states some already known facts of reservoir operation, and the functionality of reservoir in mitigating droughts. One argument authors drawn was on reservoir has capability of mitigating short-term drought, and has limited capability for multi-year droughts. This is not a surprise and it is due to the sizing of reservoir and local hydrology. None of the reservoir in the world will have unlimited resiliency for extreme water supply conditions. What is the uniqueness of the selected reservoir in Chile? and what the novelty of using author proposed methods to identify something already known?

10. Support of Conclusion: Last but most importantly, authors started the manuscript with a very new terminology of "anthropogenic drought". This is true for deforestation, urbanization, and ground water over drawing or human actives induced temperature/CO2 increases. However, this manuscript focuses on analyzing the modelled and observed data prior and after a single reservoir built in Chile. The scope of work falls better into reservoir operation, and drought mitigation by reservoir releasing strategies, instead of anthropogenic activity induced drought conditions.

11. Some English grammar issues such as:

Line 12: which basin? Line 76, "It is currently unclear on what is the . . ..", the word "on" seems to be extra. Line 389, "including a having of average duration and deficit volumes with the presence of the dam" is not clear and need to be restructured. Line 500, "in which it they have", the word "it" seems to be extra

In closing, reviewer cannot agree this manuscript been published given the fact that the current content suffers from many major issues, including the motivation, design of experiments, justification of results, support of conclusions, clarity and presentation, as reviewer listed above.

---

## Referee Comment (RC2) · Anonymous Referee #2 · 19 Dec 2016

The manuscript aims to assess the effects dam building on hydrological droughts in a case study of the Santa Juana dam in Chile. This is done by establishing liner relationships (% change) between upstream and downstream sites of the dam based on observed and modelled data (with natural and human influenced scenarios).

In its current state, I cannot recommend the publication of the manuscript, as the methods applied and the analysis conducted is not clear and appears to lack methodological rigor and the interdisciplinary aspect and dynamic interactions (here the interactions between humans (dam building activity) and the hydrological system (with regard to drought characteristics)) are not well modelled/analysed (as specified in the comments below).

General comments:

[Figure]

Overall I'm not sure if the article fits the interdisciplinary scope of the journal which focuses on the interaction in the earth system. As it is written currently the manuscript would better fit a journal focused on hydrology. This is of particular concern as at several instances in the manuscript a solid background knowledge in drought hydrology is required to understand the statements, which cannot be expected from the interdisciplinary audience of this journal (see also some of the specific comments below). I therefore encourage the authors to focus more on the journal and its audience when revising the manuscript.

Additionally, I would encourage the authors to better streamline the research presented, as the train of thought is not clear. Currently, throughout the manuscript several new aspects are being invoked in the middle and the end of the paper that should have been already considered and presented earlier on (e.g. sensitivity analysis).

Apart from these aspects, my main concern is the use of that parts of the analysis are performed on the output (streamflow) of model that is supposed simulate human influence in the catchment, although the model under the 'human influence' scenario is not simulating the river discharge correctly (L 439). Which calls this part of the research into question. Additionally, the authors do not provide any information on the WEAP model setup (e.g. reservoir operation rules, water abstractions, etc.) and sensitivities to initial values/parameters, which makes it impossible to reproduce of the research and draw any conclusions on the possible influence of the model setup on the results obtained.

To assess the influences of human on droughts a detailed analysis of the reservoir management rules and water allocations is needed, as it is not just the presence of the built structure itself that has an effect on drought but predominately how the water is managed. This very important aspect is hidden in the WAEP model and missing from the analysis and not well covered in the discussion.

In addition, the 'upstream-downstream approach' presented in the method (i.e. the

direct comparison of changes in selected drought indices upstream and downstream of the dam) is indicated as a 'new method for quantifying change' (L 144). I agree with the authors that such a 'upstream-downstream' comparison is useful in trying to determine the influence of a dam/reservoir, however, I think this does not qualify for the label 'new method'.

Additionally, I have some concerns how the comparison is being implemented in this case study. The additional catchment area between two hydrological stations that are being compared is 500 km2 (which is about 1/3 of the entire downstream catchment). With such a large additional catchment area contributing to the 'downstream part', the drought characteristics that can be evaluated at the two stations might already being naturally altered by the hydrological processes occurring (as opposed to measurements that are directly taken upstream and downstream of the structure). The presence of the large additional contribution areas makes it difficult to compare the changes in a (temporally) lumped setting, as there is a lot of scope for non-linear hydrological processes occurring. This issue needs particular attention, as the pre- and post-dam periods have different length and therefore might have experienced very different hydrological settings. Without a thorough assessment of the temporal stability of the differences between upstream and downstream through for example a sensitivity analysis of % change downstream for different temporal windows (during the pre-dam period) verifying temporal stability the current methodology is not very meaningful.

Finally, the paper is missing a detailed discussion section. In the individual paragraphs some discussion is provided, however for clarity I suggest moving these scattered parts into one single coherent discussion section.

Specific comments:

Title: Please be more specific and either replace 'in an arid region' with the study region or add the study area to the title after a colon.

L2: Please clarify why 'increased pressure on water resources' 'lead to INCREASED

management'. Consider rephrasing.

L13-15: Many of the 'findings' described in the abstract seem to be obvious. E.g. a delay in the timing of the drought with the presence of the dam. Please make sure to describe the findings more concrete.

L52: 'human activities ... can positively affect the hydrological system'. I think I know what the authors try to convey but I'm not sure if 'the histological system' is the right wording. Please rephrase.

L 53: Please define what 'resilience' means in this setting. This is would also be beneficial in L65.

L 70-72; 'population changes'. Please be more specific what these changes entail. This also applies to 'changes in supply' and 'alterations in precipitation patterns'.

L74: replace 'impacting' with 'affecting'

L 75: 'vulnerable areas' in what sense? Please elaborate.

L 76: Please elaborate WHY 'it is currently unclear on what is the best method' (what are the difficulties, why has this not been resolved so far) instead of just stating that it is unclear.

L 77: 'these research gaps' . Please elaborate which ones.

L 80: to avoid confusion I suggest stating that the dam was 'operating in 1998' and not 'built by 1998'.

L99: Figure 1: The labels in the elevation map is not readable as it is too small and blurred.

L 105 'the precipitation is inter-annually variable'. Please consider re-wording. This also applies to L 109 'most vulnerable'.

L 116: To make it easier for the reader to interpret this plot I suggest adding either

labels to the Figure marking the seasons or adding the details seasons to the figure caption. Additionally, please specify which variable is shown with the lines and the bars.

L 124: Please change the word order to 'the main water regulating structure'.

L 131: Please elaborate how the 'partial failure' and failure thresholds were established/do entail.

L 133: Before final submission, please make sure that quality of the figure is higher, as currently the labels are blurred

L 142: replace 'about' with 'of'.

L 159-160: please do not just state that missing data was interpolated and replaced with zeros together with the reference, but also elaborate for the interdisciplinary readership why this can be done in this setting and how this might/might not influence the results of the study.

L 168: add space between ')and'

L 181 'This method is based on the observation-modelling framework...' Please briefly elaborate what this framework entails.

L 186: replace 'integrating' with 'integrated'.

L 186-194: A detailed account of the WEAP model setup is needed. Please elaborate with at least one additional full paragraph.

L 190: Please reword, an objective functions does not assess the 'accuracy'. Additionally, please give the formula that was used to calculate the NSE, as log values are mentioned and the original Nash Sutcliffe efiňȦciency (NSE) does referenced does not allow for this. Please elaborate for the interdisciplinary readership why log values were used. Additionally, I strongly recommend not only relying on the NSE (which is not very sensitive to systematic model over- or underprediction, which could have a strong

influence on the drought indices derived), but also using additional model evaluation measures to quantify absolute or relative volume errors. See also Krause et al 2005. Please also elaborate why monthly data was used to assess the model performance.

Section 3.3.1. & 3.3.2: The first parts of the sections are generally written in the style of a literature review stating recommendations based on previous studies. However, often it remains unclear WHY other studies recommend a certain choice of threshold or index. I would recommend cutting these sections and focusing only on the parts that are relevant for this study and explaining why certain choice were made. Equations 1-3: All equations rely on the assumption of temporal stability (see also general comment above). Please add assessment.

L 303 - 308 the name given ('Exphum') to the variable that is supposed to represent the 'expected 'natural' value' is confusing. To avoid confusion I suggest naming the variable 'Expnat'.

L 308 & L 317: please use different symbols/names to distinguish the different ways how the '% of human influence' was calculated. Additionally, please make clear in the text what the difference between the two equations is.

L 330: Please specify/quantify the result instead of just stating that 'drought events APPEARED to be reduced'

L 331: '. . .similar meteorological droughts occurred. . .' which SPI index are you refereeing to? Please specify.

L 341: I think 'maximum duration' should read 'number of drought events' with the associated correction of the following numbers? As this is the only variable in the table that increased with the in the calculation of the % of human influence.

L 355: Do you mean 'discrepancies' instead of 'discretions'?

L 362: 'blocky pattern' does refer to which SPI? SPI-12? Please indicate in text.

L 362-365: I'm not sure why this is presented here in the results section. This should either be taken into account beforehand or moved into the discussion section.

L372: Figure 6: Panel a) is blurred. Please provide a higher resolution. Additionally, please add note to legend that the high values on the y-axis are cut.

L 381: why is it here now the normal NSE and not the log value as presented on the method section? This is confusing.

L 381-384: Please provide a Figure showing the observed and the simulated discharge to complement the NSE statistics before analysing the series for droughts.

L 401: This is not visible from Figure 7. Please provide some way how the reader follow this reasoning (e.g. summary statistics on the seasonal distribution of droughts)

L 410: the modifications in the hydrological regime are not just cause by the presence of a dam/reservoir but also strongly depend on the management/operation rules! Please also add this to the discussions

L 418: Discharge is given in mm/months in Figure 7. If this measure is used (as compared to m3/s) additional information is needed to be able to know the flow volume. Please add. (This also applies to Table 1- 3 where discharge is given in mm/d) Additionally, I'm not sure if I understand this correct but I think Figure 7 shows the 'downstream site'. If this is the case, please make clear that the reader understands this. If not, please make sure that it is clear what the figure is showing.

L 465-466: Why not compare SI and TL directly? I think this is an important part to understand which method is most appropriate for analysing the human influence on drought. I therefore suggest adding a section in this.

Section 4.6 (529-573): I'm not sure if this section detailing the effect of the dam on specific drought events is appropriate as the WEAP model performance is poor. Without any details on the models setup any results presented could be artifacts of the rules established in the WEAP model.

References: Although I recognise that one of the authors has published several papers on droughts I think the the that the self-citations are too prominent in the text, particularly in the introduction section (8 self-cites were the authors is among the first 2 authors) but also in the rest of the paper. I would encourage the authors to strive for a more balanced review of the literature/methods particularly with regard to publications outside of the authors network.

Tables: Please format the tables according to the publisher guidelines: 'Horizontal lines should normally only appear above and below the table, and as a separator between the head and the main body of the table. Vertical lines and shading must be avoided'. Additionally, please check with the typesetting if the symbols used in Table 5-8 can be used in the typesetting process.

Reference used in review: Krause, P., Boyle, D. & Bäse, F. Comparison of different efficiency criteria for hydrological model assessment. Advances in Geosciences 5, 89–97 (2005).

---

## Author Comment (AC1) · 10 Feb 2017

*We thank reviewer for their comments and insight into this manuscript. We have addressed every point that they have made, with our responses below in blue italics.*

Anonymous Referee #1

This paper studies the anthropogenic effect of the Santa Juana dam on drought in the Huasco basin in Chile. The Standardized Indices (SI) and Threshold Level (TL) method with different thresholds are applied to observed and modelled data. The study illustrates the effect of the Santa Juana dam on different characteristics of the drought, including duration and intensity. Also, the performance of the SI and TL method on the observed and simulated data is compared. The topic of this manuscript is interesting, as reviewer found the study was more like to investigate the reservoir operation and reservoir's capability on mitigating drought conditions instead of drought monitoring and prediction. The current manuscript suffers from some major issues as reviewer listed below. Major comments:

1. Motivation: Reservoirs are able to change the patterns and magnitudes of streamflow. This is an unknown fact. Reviewer is unclear about the experiment design of this study. If both observed and simulated results shown reservoir was able to delay the timing, frequency or magnitude of drought events, then this again is a restate of known fact. Therefore, how this study can be used for more efficient reservoir operation or planning for droughts events still have questions. Authors are suggested to summarize more about recent published paper in WRR and GRL to better formulate the motivation of this study. (search key words: reservoir operation, water allocation, drought conditions).

*We agree that the motivation behind the paper could be clearer, and we propose changes to the paper to address this, including changing the title, refining the abstract and the introduction, as well as minor changes to the methods section, major changes to the results section, and the inclusion of a discussion section.*

*The focus of the paper is: Finding the best method for quantifying the human influence on hydrological drought using observation data, by 1) comparing the threshold level method and standardised indices; and 2) testing the upstream-downstream approach to quantify the human influence from discharge data (in which we refer to in the paper – using discharge stations upstream and downstream of a human activity).*

*The specific reservoir is not the focus, but rather how these methods can be applied to quantify the impact of a human activity. By reorientation of the focus from Chile and the reservoir and those results, we consider now the methodological aspect of quantifying the human influence, and the sensitivity analysis of it. The new title reflects this re-focus:*

***"Evaluating observation-based methods to quantify human influences on hydrological droughts"***

*Suggested keywords: drought, analysis methods, quantification, anthropogenic activities*

*Proposed revised manuscript headings and subheadings:*

*1   **Introduction***

2. Motivation and Organization: The introduction has described why drought is important and previous studies focusing on the relationship between human activities and drought conditions. However, reviewer find it difficult to understand the contribution of this study with respect to those mentioned studies in this manuscript. What is the scientific problem that this study is trying to solve and **how this study will contribute to those existing findings about human activities amplified or alleviated drought condition?** Reviewer believes this study is not targeting on methodology advances rather than application of certain existing analytical methods to a case study in Chile. Then, the question becomes how this application is unique and novel as compared to the uses of similar techniques to analyze the functionality of reservoirs on droughts.

*We stress that this study contributes significantly to the problems outlined in the introduction of this paper by looking at the best method for quantifying the human influence on hydrological droughts through the use of Chilean data as an exemplar. Tools are of crucial importance for increasing the understanding of droughts in the Anthropocene. The major findings from this study are twofold:*

*1) Identifying the most appropriate drought analysis method for quantifying the human influence using observation data. This is something which is novel from an observation based approach;*

*2) A sensitivity analysis between two commonly applied drought analysis methods, standardised indices and threshold level, enabling recommendations on the best method to use to identify the anthropogenic influences on hydrological droughts.*

3. Justification of results: In results section (line 337-338), authors also quoted that "This temporal difference between observed droughts upstream and downstream reflects the impact of human activities, also observed in other studies (Assani et al., 2013; Liu et al., 2106)." Then, the explanation is needed to justify in what aspects this study will differ from others?

*We have stated that our results show that timing of droughts have been affected by human activities, which we have managed to find in limited studies elsewhere. But we would like to*

*emphasize (and this will be done in the revised version of the paper) that here we are actually producing quantification for the changes caused by an anthropogenic activity. It is this quantification aspect which is especially novel for the research area.*

*Furthermore, this paper itself is testing the application of the upstream-downstream approach for this research area and purpose, which is different to the existing observation-modelling framework and simulated naturalised discharge approaches because it only uses observation data. The quantification of the human influence through the upstream-downstream approach is explained in the paper (with the equations) so that this proposed methodology can be replicated with other case studies.*

4. Justification of results: The "human-influenced" results only reached NSE=0.454, which is a rather poor performance. Considering the temporal resolution (monthly) and the length of analysis (over decade), this NSE value lacks proper accuracy to represent any simulation of human activities. This critically undermines the main motivation of analyzing the functionalities of reservoirs in alleviating drought conditions.

*We agree that the human-influenced WEAP model scenario NSE results are low, as stated in the paper. However, we mention an important reason as to why this might be the case in the paper: because the model may not be accurately modelling human responses to drought situations (e.g. increase in water use, decrease in water use due to restrictions etc).*

*With our changes, we are proposing to take the focus away from the modelling aspect of this paper, by removing the modelling data part, refine the focus, and helping us to address a number of comments made by both reviewers.*

5. Justification of results: line 15, "A delay in timing of drought events has been observed also with the presence of the dam" how is this being illustrated in the results and conclusion as this is one of the main concluding remarks in the end?

*This is already show visually by the figures in the results section. However, in the revised version we have analysed and added a summary of the seasonal distribution of droughts into the results section (3.1 Results of human influence on drought characteristics downstream), as suggested by Reviewer 2. These results illustrate the importance of dam impoundment and reservoir management (human influences) on the timing of hydrological droughts downstream.*

6. Experiment Design: Reviewer also found it is confusing about the experimental design. For instance, in Figure 4, both daily data from 1965-2013 for upstream discharge (this becomes reservoir's inflow) and downstream discharge (reservoir's outflow) were used in the "pre-dam versus post-dam" scenario. The results for the first scenario was compared with a "naturalized versus Human-influenced" scenario as shown on the right panel, in which the resolution of data become monthly and coverage of data starts from 1960 to 2010. It is confusing, or lack of explanation why different resolution and lengths of data were used and compared. Isn't that Pre-dam the same as Naturalized, and Post-dam is human-influenced?

*It is important to highlight that the time periods between the observation and modelled data used in the paper do not align directly because it is based on using available data, which can be especially limiting in an arid, developing country. The observation data runs 1965 – 2013 due to missing data before 1965. The WEAP model was already set up for the basin and had data from 1960 – 2010. However, with the emphasis now away from the WEAP model, this difference in time periods should be less problematic as all the observation data compared in the revised version is now consistent: 1965 – 2013.*

*We would also like to stress that for the sensitivity analysis we have kept everything consistent– such as the time period, the temporal resolution (monthly), the threshold (TL using variable $80^{th}$ percentile and SI using -0.8) and using the whole time period for the threshold in both SI and TL. The sensitivity analysis discussion also highlights the potential difference for using daily and monthly data. This sensitivity analysis has now been moved forward in the paper to being one of the key results (section 3.2).*

*With the new changes, we now just look at the pre-dam upstream and downstream to establish the relationship in the undisturbed period, and we use the post-dam upstream as the "natural situation" and the post-dam downstream as the "human-influenced situation" to enable a comparison in the same time to ensure the same external factors (e.g. meteorological).*

7. Experiment Design: Reviewer noticed that the length of the observed data and simulated data are different (Figure 5, 6, and 7). Any comments on some drought events happened after 2010 as many figures are showing a significant decreasing trend during the recent years?

*Please see our response to the point above in regards to the different time periods. We comment on the change in drought frequencies between the upstream and the downstream stations, with a decrease in events observed downstream during the recent years in comparison to upstream, due to the reservoir.*

8. Clarity and Presentation: A general overview of the WEAP model (inputs, output, general structure. . .) can give better understanding of the result section, as the author mentioned variation in water use as the reason for the performance of the model.

*We propose to remove the information and results from the WEAP model element and focus on the observation data and the two methods (ST and TL).*

9. Support of Conclusion: Line 575-592. This section again states some already known facts of reservoir operation, and the functionality of reservoir in mitigating droughts. One argument authors have drawn was on reservoir has capability of mitigating short-term drought, and has limited capability for multi-year droughts. This is not a surprise and it is due to the sizing of reservoir and local hydrology. None of the reservoir in the world will have unlimited resiliency for extreme water supply conditions. What is the uniqueness of the selected reservoir in Chile? And what the novelty of using author proposed methods to identify something already known?

*The paper is not about the reservoir itself, but the method and idea of quantifying the human influence. Therefore, the method is not only applicable to the Santa Juana reservoir in Chile, but to other reservoirs and other human activities across the world.*

10. Support of Conclusion: Last but most importantly, authors started the manuscript with a very new terminology of "anthropogenic drought". This is true for deforestation, urbanization, and ground water over drawing or human actives induced temperature/CO2 increases. However, this manuscript focuses on analyzing the modelled and observed data prior and after a single reservoir built in Chile. The scope of work falls better into reservoir operation, and drought mitigation by reservoir releasing strategies, instead of anthropogenic activity induced drought conditions.

*We do not use the term "anthropogenic drought" in the paper, we refer to anthropogenic activities and how they influence drought. These influences could be negative (drought aggravation) or positive (drought alleviation, e.g. like the reservoir analysed here). We argue that the human actions of building and managing a reservoir is very much a direct anthropogenic activity that changes the severity and timing of drought compared to the normal, as seen here. In our study the reservoir had an alleviating effect on drought conditions, but different types of reservoirs can have negative impacts (aggravating), e.g. reservoirs built and managed for the purpose of supplying drinking water rather than providing water security downstream.*

*We disagree with the reviewers comment about the scope of the work and put this down to a misinterpretation of our aims, which we have now clarified (see aims). Furthermore, in changing the title and adding in sentences in the introduction, we believe that we have framed the scope of the work better.*

11. Some English grammar issues such as: Line 12: which basin? Line 76, "It is currently unclear on what is the . . ..", the word "on" seems to be extra. Line 389, "including a having of average duration and deficit volumes with the presence of the dam" is not clear and need to be restructured. Line 500, "in which it they have", the word "it" seems to be extra.

*Thank you for highlighting these grammar errors, we made changes.*

---

## Author Comment (AC2) · 10 Feb 2017

*We thank the reviewer for the time and effort with the review. We have addressed every point that they have made, with our responses below in blue italics.*

Anonymous Referee #2

The manuscript aims to assess the effects dam building on hydrological droughts in a case study of the Santa Juana dam in Chile. This is done by establishing liner relationships (% change) between upstream and downstream sites of the dam based on observed and modelled data (with natural and human influenced scenarios).

In its current state, I cannot recommend the publication of the manuscript, as the methods applied and the analysis conducted is not clear and appears to lack methodological rigor and the interdisciplinary aspect and dynamic interactions (here the interactions between humans (dam building activity) and the hydrological system (with regard to drought characteristics)) are not well modelled/analysed (as specified in the comments below).

General comments:

Overall I'm not sure if the article fits the interdisciplinary scope of the journal which focuses on the interaction in the earth system. As it is written currently the manuscript would better fit a journal focused on hydrology. This is of particular concern as at several instances in the manuscript a solid background knowledge in drought hydrology is required to understand the statements, which cannot be expected from the interdisciplinary audience of this journal (see also some of the specific comments below). I therefore encourage the authors to focus more on the journal and its audience when revising the manuscript.

*We hope that you find that we have addressed a lot of your comments, mainly through a refocusing and restructuring of the paper (please see new proposed title and headings and sub-heading).*

*"Evaluating observation-based methods to quantify human influences on hydrological droughts"*

*1 Introduction*
*2 Methods*
*2.1 Observation data*
*2.2 Drought analysis approaches*
 *2.2.1 Threshold level*
 *2.2.2 Standardised Indices*
*2.3 Sensitivity analysis*
*2.4 Estimation of the human impact on drought characteristics*
 *2.4.1 Percentage change due to human influence in observation data*
*3 Results*
 *3.1 Results of human influence on drought characteristics downstream*
 *3.2 Sensitivity analysis of drought analysis methods (SI vs TL)*
*4 Discussion*
 *4.1 ST vs TL*
 *3.2 Comparison between observed data and modelled data*

**4   Conclusions**

*We deem the subject of this paper, the anthropogenic influence on hydrology and droughts, to be an important interdisciplinary geoscience problem, making it appropriate for Earth System Dynamics. The research topic fits into the broader theme of "interactions between human and "natural" processes in the Earth System".*

*We will change some of the terminology and writing to be more appropriate to an interdisciplinary audience.*

Additionally, I would encourage the authors to better streamline the research presented, as the train of thought is not clear. Currently, throughout the manuscript several new aspects are being invoked in the middle and the end of the paper that should have been already considered and presented earlier on (e.g. sensitivity analysis).

*We propose to rewrite a number of parts to the paper and restructure it so that the focus is clearer, including the title, the abstract and the results (please see revised headings and sub-headings to demonstrate this).*

*We also agree that the sensitivity analysis could be moved forward in the paper to hold more importance to the results and the paper itself and the results section now focuses on 1) reporting briefly the observation results using the SI and TL methods; 2) comparison between the two methods – the sensitivity analysis.*

Apart from these aspects, my main concern is the use of that parts of the analysis are performed on the output (streamflow) of model that is supposed simulate human influence in the catchment, although the model under the 'human influence' scenario is not simulating the river discharge correctly (L 439). Which calls this part of the research into question. Additionally, the authors do not provide any information on the WEAP model setup (e.g. reservoir operation rules, water abstractions, etc.) and sensitivities to initial values/parameters, which makes it impossible to reproduce of the research and draw any conclusions on the possible influence of the model setup on the results obtained.

*We agree that there is more information needed about the WEAP model set up itself, a comment also made by reviewer 1. However, we have completely removed the WEAP aspect of the paper.*

To assess the influences of human on droughts a detailed analysis of the reservoir management rules and water allocations is needed, as it is not just the presence of the built structure itself that has an effect on drought but predominately how the water is managed. This very important aspect is hidden in the WAEP model and missing from the analysis and not well covered in the discussion.

*We would like to stress that reservoir management is not the focus of the study. We completely agree that the reservoir management rules and water allocations are needed for a more detailed analysis; however, what we can provide here with the data is an initial assessment and quantification of how the presence of that reservoir is influencing hydrological droughts downstream compared to those observed upstream in the same time period.*

*Using observations from the pre-dam period, we can also estimate how much the reservoir has changed droughts from what could be expected at the downstream station based on upstream observations. By using observation data the management of the reservoir is included by comparing the upstream and downstream stations of the reservoir.*

In addition, the 'upstream-downstream approach' presented in the method (i.e. the direct comparison of changes in selected drought indices upstream and downstream of the dam) is indicated as a 'new method for quantifying change' (L 144). I agree with the authors that such a 'upstream-downstream' comparison is useful in trying to determine the influence of a dam/reservoir, however, I think this does not qualify for the label 'new method'.

*In the paper we make a new application of this method, to quantifying the human influence for hydrological droughts. We will look to change the label 'new method' to 'new application' of this method for droughts.*

Additionally, I have some concerns how the comparison is being implemented in this case study. The additional catchment area between two hydrological stations that are being compared is 500 km2 (which is about 1/3 of the entire downstream catchment). With such a large additional catchment area contributing to the 'downstream part', the drought characteristics that can be evaluated at the two stations might already being naturally altered by the hydrological processes occurring (as opposed to measurements that are directly taken upstream and downstream of the structure). The presence of the large additional contribution areas makes it difficult to compare the changes in a (temporally) lumped setting, as there is a lot of scope for non-linear hydrological processes occurring. This issue needs particular attention, as the pre- and post-dam periods have different length and therefore might have experienced very different hydrological settings. Without a thorough assessment of the temporal stability of the differences between upstream and downstream through for example a sensitivity analysis of % change downstream for different temporal windows (during the pre-dam period) verifying temporal stability the current methodology is not very meaningful.

*We agree that our approach does not take into account a possible non-linear relationship between the upstream and downstream site, however it is a preliminary assessment of the propagation, and is more accurate than not accounting for it. We do also present the results of the percentage change downstream if readers do not agree with our method.*

*Furthermore, this proposed method is better than when pre-dam time periods are compared directly with post-dam time periods which have completely different driving external forcing of meteorological conditions.*

*However we will add this limitation in the paper: that we are using the assumption of a linear relationship between upstream and downstream stations to estimate the human influence.*

*It is true that the pre-dam and post-dam periods have different record lengths, and it is for this reason that we standardised the number of droughts to be per decade rather than the absolute number (record length influences the value), but record length does not affect the values reported for duration and deficit.*

*We conducting the extra analysis suggested by the reviewer (different temporal windows during the pre-dam period), however this is a difficult task based on the fact that droughts span over multiple years, and a slight delay between droughts can then make them disassociated when occurring in two adjacent time windows.*

*Instead, we directly compared a number of drought events in the pre- and post-dam period (table below). We can verify that the changes observed in individual drought events during the post-dam period are much larger than those observed in the pre-dam period, implying that the changes measured are outside of the range of noise and uncertainty. Most of the numbers of the pre-dam period are in the same range, which gives us confidence in using this approach. Although we think it is better to do this analysis on averages than on individual events, the fact that the post-dam numbers are substantially higher indicates that the effect of the dam is overruling the propagation effect.*

| | Sample drought event | Start date upstream | Start date downstream | % change downstream | |
|---|---|---|---|---|---|
| | | | | Duration | Deficit |
| *Pre-dam* | Drought 1 | 08/1968 | 08/1968 | +1.5% | +28% |
| | Drought 2 | 08/1975 | 09/1975 | -18% | -30% |
| | Drought 3 | 08/1979 | 11/1979 | -26% | -10% |
| | Drought 4 | 11/1990 | 11/1990 | -5% | -35% |
| *Post-dam* | Drought 5 | 09/2009 | 05/2010 | -79% | -90% |
| | Drought 6 | 09/2012 | 01/2013 | -81% | -98% |

Finally, the paper is missing a detailed discussion section. In the individual paragraphs some discussion is provided, however for clarity I suggest moving these scattered parts into one single coherent discussion section.

*We agree that the paper as it stands does not have a standalone discussion section. We had decided to weave our discussion points in with the results section given that number of different aspects that are presented in the results section.*

*However, we have now restructured the paper to focus on the results and then the discussion separately.*

*We thank the reviewer for their specific comments, and we have changed them or addressed all individually. Some of these may become irrelevant with the new restructuring of the paper.*

Specific comments: Title: Please be more specific and either replace 'in an arid region' with the study region or add the study area to the title after a colon.

*New title: **"Evaluating observation-based methods to quantify human influences on hydrological droughts"***

L2: Please clarify why 'increased pressure on water resources' 'lead to INCREASED management'. Consider rephrasing.

*We have edited the sentence to now read* "Human activities affecting hydrology are increasing in occurrence with growing pressure on water resources and availability, however, the impacts of these anthropogenic activities on hydrological droughts have yet to be incorporated and assessed."

L13-15: Many of the 'findings' described in the abstract seem to be obvious. E.g. a delay in the timing of the drought with the presence of the dam. Please make sure to describe the findings more concrete.

*The delay in timing of droughts with the presence of the dam are not stated so much in the literature, and our results are the quantitative assessment of the changes.*

*However, we have re-written our abstract and the focus is now removed from the reservoir results, therefore this comment should not be an issue anymore.*

L52: 'human activities . . . can positively affect the hydrological system'. I think I know what the authors try to convey but I'm not sure if 'the histological system' is the right wording. Please rephrase.

*Thank you for picking this up, we have changed this sentence to say* "water availability" *instead of* "the hydrological system".

L 53: Please define what 'resilience' means in this setting. This is would also be beneficial in L65.

*We define the term resilience as* "the ability of a system to persist in a given state subject to perturbations" *(Folke et al., 2010) to represent the ability of the community to cope with changes. However, we have rewritten this section and the term resilience is not present anymore.*

L 70-72; 'population changes'. Please be more specific what these changes entail. This also applies to 'changes in supply' and 'alterations in precipitation patterns'.

*"Population changes" includes population increase and westernisation of lifestyles. "Changes in supply" is already explained by the rest of the sentence implying that changes in temperature and precipitation will affect water availability. Alterations in precipitation patterns refers to the fact that not only is precipitation predicted to decrease and increase with climate change, but the timing and intensity of precipitation is expected to change.*

L74: replace 'impacting' with 'affecting'

*Changed.*

L 75: 'vulnerable areas' in what sense? Please elaborate.

*"Vulnerable areas" is used here because we have already stated what factors are affecting these regions in regards to their water availability (increase in demand and changes to supply) and that they have low resilience. However, once again in the rewriting, we have removed this sentence.*

L 76: Please elaborate WHY 'it is currently unclear on what is the best method' (what are the difficulties, why has this not been resolved so far) instead of just stating that it is unclear.

*We have addressed this in the revised version.*

L 77: 'these research gaps' . Please elaborate which ones.

*The research gaps are identified in the two sentences prior: "Therefore, there is a need to improve our knowledge on how human activities are impacting on drought to enable better drought preparation and mitigation, especially in these vulnerable, arid regions. It is currently unclear what is the best method for assessing and quantifying the impact of human activities on hydrological droughts."*

*We have re-written this section and we have specifically highlighted the research gaps before mentioning how the paper aims to address them.*

L 80: to avoid confusion I suggest stating that the dam was 'operating in 1998' and not 'built by 1998'.

*Changed.*

L99: Figure 1: The labels in the elevation map are not readable as it is too small and blurred.

*Changed.*

L 105 'the precipitation is inter-annually variable'. Please consider re-wording. This also applies to L 109 'most vulnerable'.

*"the precipitation is inter-annually variable" has been changed to "the precipitation can vary from year to year with the ENSO".*

*"Most vulnerable" has been changed to "most susceptible" .*

L 116: To make it easier for the reader to interpret this plot I suggest adding either labels to the Figure marking the seasons or adding the details seasons to the figure caption. Additionally, please specify which variable is shown with the lines and the bars.

*We agree and have changed the caption to read: "**Figure 2:** Seasonality plots for monthly precipitation bar charts and overlaid discharge line graph using daily data (1965-2013)." And we can add on vertical dashed lines framing the Chilean winter (May and August) and stating "Winter" on that space on the graph.*

*This figure has also been removed from the main manuscript and is in the supplementary material instead.*

L 124: Please change the word order to 'the main water regulating structure'.

*Changed.*

L 131: Please elaborate how the 'partial failure' and failure thresholds were established/do entail.

*Changed to: "During the recent multi-year drought (2007-2015), by 2011 reservoir levels dropped below levels of "partial failure" (<100 Mm³, Figure 3), levels set by the Huasco River Supervisory Board, resulting in them implementing severe water restrictions (through its operational model)."*

L 133: Before final submission, please make sure that quality of the figure is higher, as currently the labels are blurred

*Changed.*

L 142: replace 'about' with 'of'.

*Changed.*

L 159-160: please do not just state that missing data was interpolated and replaced with zeros together with the reference, but also elaborate for the interdisciplinary readership why this can be done in this setting and how this might/might not influence the results of the study.

*Changed.*

L 168: add space between ')and'

*Changed*

L 181 'This method is based on the observation-modelling framework. . .' Please briefly elaborate what this framework entails.

*We can easily add a sentence or two introducing this framework. This is now in the introduction when the research gaps are highlighted: we include more about existing methods such as the observation-modelling framework and how this paper tests a more observation based approach, the upstream-downstream approach.*

L 186: replace 'integrating' with 'integrated'.

*Changed.*

L 186-194: A detailed account of the WEAP model setup is needed. Please elaborate with at least one additional full paragraph.

*We have completely removed the WEAP modelling aspect so this should not be a problem now.*

L 190: Please reword, an objective functions does not assess the 'accuracy'. Additionally, please give the formula that was used to calculate the NSE, as log values are mentioned and the original Nash Sutcliffe efficiency (NSE) does referenced does not allow for this. Please elaborate for the interdisciplinary readership why log values were used. Additionally, I strongly recommend not only relying on the NSE (which is not very sensitive to systematic model over- or under-prediction, which could have a strong influence on the drought indices derived), but also using additional model

evaluation measures to quantify absolute or relative volume errors. See also Krause et al 2005. Please also elaborate why monthly data was used to assess the model performance.

*The NSE values were indeed logged, however now that the WEAP model has been removed; this is no longer an issue.*

Section 3.3.1. & 3.3.2: The first parts of the sections are generally written in the style of a literature review stating recommendations based on previous studies. However, often it remains unclear WHY other studies recommend a certain choice of threshold or index. I would recommend cutting these sections and focusing only on the parts that are relevant for this study and explaining why certain choices were made. Equations 1- 3: All equations rely on the assumption of temporal stability (see also general comment above). Please add assessment.

*Whilst we agree that these sections read as literature reviews, we think that it is important to have this background information in for the interdisciplinary audience, so that information is not lost when showing the results from the two different methods. These sections now hold more value as the focus of the paper is on comparing the two methods.*

*Equations 1-3 do not require temporal stability because we are directly comparing the drought events of two situations which look to use the same time period and the same inputs (e.g. upstream and downstream).*

L 303 - 308 the name given ('Exp-hum') to the variable that is supposed to represent the 'expected 'natural' value' is confusing. To avoid confusion I suggest naming the variable 'Exp-nat'.

*We agree that these abbreviations are confusing and we have indeed re-named: "This generates an expected "natural" value (Exp$_{-down}$) for the post-dam period which could be directly compared to the actual Qobs$_{-down}$ post-dam value. The difference between this expected value (Exp$_{-down}$) and the actual observed value (Obs$_{-hum}$) gave an overall percentage of human influence in the post-dam period (Eq 2)."*

L 308 & L 317: please use different symbols/names to distinguish the different ways how the '% of human influence' was calculated. Additionally, please make clear in the text what the difference between the two equations is.

*We have changed the name of equation 2:* **Percentage of human influence (observations) (%)**

*Equation 3 has been removed as it was linked to the modelling data.*

L 330: Please specify/quantify the result instead of just stating that 'drought events APPEARED to be reduced'

*We have re-written so that the quantified results are outlined briefly as the results sub-section from the data itself.*

L 331: '. . .similar meteorological droughts occurred. . .' which SPI index are you refereeing to? Please specify.

*We can change this to be specific to the SPI we are talking about, SPI -6.*

L 341: I think 'maximum duration' should read 'number of drought events' with the associated correction of the following numbers? As this is the only variable in the table that increased with the in the calculation of the % of human influence.

*It is correct at the moment as maximum duration, please see table 2 post-dam for the values that this sentence is referring to.*

L 355: Do you mean 'discrepancies' instead of 'discretions'?

*Changed.*

L 362: 'blocky pattern' does refer to which SPI? SPI-12? Please indicate in text.

*SPI-6 has been specified in the text now.*

L 362-365: I'm not sure why this is presented here in the results section. This should either be taken into account beforehand or moved into the discussion section.

*The paper we submitted had a combined results and discussion section, which is why this is presented here. However, we have restructured the paper and so we have moved this to later on, in a discussion section.*

L372: Figure 6: Panel a) is blurred. Please provide a higher resolution. Additionally, please add note to legend that the high values on the y-axis are cut.

*Changed.*

L 381: why is it here now the normal NSE and not the log value as presented on the method section? This is confusing.

*We did miss out the key word 'log value' here and in fact we do refer to only the log NSE values throughout, however this is now not in the rewritten version.*

L 381-384: Please provide a Figure showing the observed and the simulated discharge to complement the NSE statistics before analysing the series for droughts.

*In the version of the manuscript we submitted, we did already show these discharges in Figure 6 (observation discharge) and Figure 7 (simulated discharge). However, now that the modelled data has been removed, this should not be an issue anymore.*

L 401: This is not visible from Figure 7. Please provide some way how the reader follows this reasoning (e.g. summary statistics on the seasonal distribution of droughts)

*This would be a very good addition to the paper and we will indeed add in a summary statistics table on the seasonal distribution of droughts.*

L 410: the modifications in the hydrological regime are not just cause by the presence of a dam/reservoir but also strongly depend on the management/operation rules! Please also add this to the discussions

*This is completely correct, and different reservoir management and operation rules would lead to a different impact on droughts downstream. For this reason, we have now stressed the purpose of reservoir at the start of the paper in the introduction section, and we can add a statement into the discussion section about this.*

L 418: Discharge is given in mm/months in Figure 7. If this measure is used (as compared to m3/s) additional information is needed to be able to know the flow volume. Please add. (This also applies to Table 1- 3 where discharge is given in mm/d) Additionally, I'm not sure if I understand this correct but I think Figure 7 shows the 'downstream site'. If this is the case, please make clear that the reader understands this. If not, please make sure that it is clear what the figure is showing.

*We can indeed add the m3/s discharge values as well as the mm/d or mm/month if needed. However, the catchment areas are given in Table 1 and by using mm/d or mm/month, the values are more comparable. Table 1 has actually been removed in the rewrite as the data is described in the start of the methods and the table is not needed.*

*You are correct about Figure 7 showing only the downstream station and this has now been clarified in the figure caption, however this figure is not in the revised version that we have been working on.*

L 465-466: Why not compare SI and TL directly? I think this is an important part to understand which method is most appropriate for analysing the human influence on drought. I therefore suggest adding a section in this.

*We do compare SI and TL methods, showing the results from both and the sensitivity analysis. They are fundamentally different methods, and therefore differences are seen between them, because one involves using the whole period to establish its threshold (SI) and the other uses a reference period to help remove the human influenced data from the threshold (TL). All of these differences are explained in Section 4.4 and section 4.5 of the submitted manuscript, and these remain in the rewritten version, with more emphasis on the comparison.*

Section 4.6 (529-573): I'm not sure if this section detailing the effect of the dam on specific drought events is appropriate as the WEAP model performance is poor. Without any details on the models setup any results presented could be artifacts of the rules established in the WEAP model.

*We agree that the use of the WEAP model is not a primary focus here because of its lower performance NSE log value. However, it does show some interesting suggestions about the 1969 drought event which could not established by the observation data: the idea that the presence of the Santa Juana reservoir will have not been able to provide much buffer again the Great Drought of 1969. We have removed the aspect of the WEAP model, and now we only refer to it to demonstrate the advantages of using modelling data, without giving concrete results because of the NSE values.*

References: Although I recognise that one of the authors has published several papers on droughts I think that the self-citations are too prominent in the text, particularly in the introduction section (8 self-cites were the authors is among the first 2 authors) but also in the rest of the paper. I would encourage the authors to strive for a more balanced review of the literature/methods particularly with regard to publications outside of the authors network.

*We look back at the paper and agree that there is a need for a more balanced reporting of the literature, and we have addressed this and will hope that the revised version would be found to be more balanced.*

Tables: Please format the tables according to the publisher guidelines: 'Horizontal lines should normally only appear above and below the table, and as a separator between the head and the main body of the table. Vertical lines and shading must be avoided'. Additionally, please check with the typesetting if the symbols used in Table 5-8 can be used in the typesetting process.

*Changed.*

Reference used in review: Krause, P., Boyle, D. & Bäse, F. Comparison of different efficiency criteria for hydrological model assessment. Advances in Geosciences 5, 89–97 (2005).